# Melanopsin (*Opn4*) is an oncogene in cutaneous melanoma

Leonardo Vinícius Monteiro de Assis [1,7,8 ✉], José Thalles Lacerda[1,8], Maria Nathália Moraes[2], Omar Alberto Domínguez-Amorocho[3], Gabriela Sarti Kinker[4], Davi Mendes [5], Matheus Molina Silva[5], Carlos Frederico Martins Menck [5], Niels Olsen Saraiva Câmara [3] & Ana Maria de Lauro Castrucci[1,6]

The search for new therapeutical targets for cutaneous melanoma and other cancers is an ongoing task. We expanded this knowledge by evaluating whether opsins, light- and thermo-sensing proteins, could display tumor-modulatory effects on melanoma cancer. Using different experimental approaches, we show that melanoma cell proliferation is slower in the absence of *Opn4*, compared to *Opn4*[WT] due to an impaired cell cycle progression and reduced melanocyte inducing transcription factor (*Mitf*) expression. In vivo tumor progression of *Opn4*[KO] cells is remarkably reduced due to slower proliferation, and higher immune system response in *Opn4*[KO] tumors. Using pharmacological assays, we demonstrate that guanylyl cyclase activity is impaired in *Opn4*[KO] cells. Evaluation of Tumor Cancer Genome Atlas (TCGA) database confirms our experimental data as reduced *MITF* and *OPN4* expression in human melanoma correlates with slower cell cycle progression and presence of immune cells in the tumor microenvironment (TME). Proteomic analyses of tumor bulk show that the reduced growth of *Opn4*[KO] tumors is associated with reduced *Mitf* signaling, higher translation of G2/M proteins, and impaired guanylyl cyclase activity. Conversely, in *Opn4*[WT] tumors increased small GTPase and an immune-suppressive TME are found. Such evidence points to OPN4 as an oncogene in melanoma, which could be pharmacologically targeted.

[1] Laboratory of Comparative Physiology of Pigmentation, Department of Physiology, Institute of Biosciences, University of São Paulo, São Paulo, Brazil. [2] Laboratory of Neurobiology, Department of Physiology and Biophysics, Institute of Biomedical Sciences, University of São Paulo, São Paulo, Brazil. [3] Laboratory of Transplantation Immunobiology, Institute of Biomedical Sciences, University of São Paulo, São Paulo, Brazil. [4] Laboratory of Translational Immuno-Oncology A. C. Camargo Cancer Center – International Research Center, São Paulo, Brazil. [5] DNA Repair Lab, Department of Microbiology, Institute of Biomedical Sciences, University of São Paulo (USP), São Paulo, Brazil. [6] Department of Biology, University of Virginia, Charlottesville, VA, USA. [7] Present address: Institute of Neurobiology, Center for Brain, Behavior, and Metabolism, University of Lübeck, Lübeck, Germany. [8] These authors contributed equally: Leonardo Vinícius Monteiro de Assis, José Thalles Lacerda. ✉email: deassis.leonardo@alumni.usp.br

Cutaneous melanoma (CM) cancer represents about 5% of all skin-related cancer cases, but it accounts for approximately 80% of cancer-related deaths. The incidence of CM has steadily increased over the years[1,2], despite increasing awareness of the deleterious effects of the sun on the skin[3]. The etiology of CM is multifactorial and includes risk factors such as UV radiation exposure, genetic susceptibility, high nevus density, reduced pigmentation, and immunosuppression[4,5].

The molecular biology of cutaneous melanoma is well understood due to an effort made by several pioneering endeavors that resulted in solid comprehension of the tumor biology landscape[6]. Currently, CM can be classified into four subtypes based on the most prevalent mutations: mutant B-Raf proto-oncogene, serine/threonine kinase (BRAF), mutant Kirsten rat sarcoma viral proto-oncogene (RAS), mutant neurofibromin 1 (NF1), and triple wild-type[6]. Intriguingly, CM is known to display the highest mutational load of all cancers[7].

The temporal control of physiological processes is crucial for homeostasis and such regulation is dependent on the circadian clock[8]. The molecular clock system is comprised of several genes that engage in transcriptional feedback loops, whose transcripts oscillate throughout the day[8–10]. In recent years, the antitumoral role of the molecular clock has been extensively investigated in several types of cancer[11–15]. In CM, expression of clock genes and proteins are mostly downregulated when compared to healthy skin or tumor-adjacent tissues in murine in vitro and in vivo, respectively, as well as in human tissues[16–21].

Melanocytes are known to be light-responsive cells mainly because they express a complex photosensitive system comprised of chromophores and light-sensitive molecules, known as opsins[22,23]. We have previously investigated the role of melanopsin (OPN4) in murine melanocytes and demonstrated that this protein acts as UVA-sensor for pigmentary and apoptosis-dependent processes[24,25] as well as a thermal sensor[26]. More recently, we demonstrated that OPN4 may also display light- and thermo-independent roles as *Opn4* knockout murine melanocytes show faster proliferation and cell cycle progression[27].

In this study, we evaluated the putative role of OPN4 in the carcinogenic process of melanoma. To this end, we investigated how the absence of OPN4 in tumor cells would affect the development and progression of melanoma in a murine model. Taken altogether, we provide evidence that OPN4 can act as an oncogene in melanoma.

## Results

**Absence of OPN4 results in slower tumor growth and increased immune cell infiltration in the tumor microenvironment (TME).** We previously established and validated an *Opn4^KO* model using Clustered Regularly Interspaced Short Palindromic Repeats (CRISPR) of B16-F10 cells[25]. In this process, three clones that exhibited no ultraviolet A radiation-induced pigmentation and apoptosis responses were identified. B16-F10 *Opn4^KO* clone 16 was chosen and Sanger sequencing of the CRISPR edited region showed alteration in the coding sequencing that led to the loss of function. Immunocytochemistry of OPN4 revealed increased protein presence in a region capping the nucleus, thus suggestive of protein retention likely due to altered protein structure[25]. In this study, B16-F10 *Opn4^KO* clone 16 was chosen and used in the next steps.

To evaluate whether OPN4 would impact tumor development, we inoculated C57Bl/6 J mice, kept in thermoneutrality (30 ± 1 °C), with B16-F10 *Opn4^WT* or B16-F10 *Opn4^KO* cells. At temperatures between 29 and 31 °C, mice do not activate thermogenesis to sustain core body temperature. In fact, mice kept below thermoneutrality are considered cold-stressed due to

increased energy requirements to sustain core body temperature[28]. Therefore, to avoid confounding factors caused by cold stress, mice were kept in their thermal-neutral temperature.

It has been previously shown that tumor growth is significantly faster in mice kept at temperatures below thermoneutrality than at thermoneutrality[29]. Indeed, tumor growth was only measurable from the 13th day onwards unlike previously shown in mice kept at 22 °C and inoculated with *Opn4^WT* cells[20]. On the 22nd and 25th days after inoculation, tumor volume was significantly smaller in *Opn4^KO* inoculated animals compared to *Opn4^WT* tumors (Fig. 1a). Tumor weight and melanin content on the 25th day were also lower in *Opn4^KO* tumors compared to *Opn4^WT* counterparts (Fig. 1b, c).

Hemogram analysis of tumor-bearing mice revealed no difference in circulating white blood cells and the absolute number of lymphocytes, but a decreased frequency of lymphocytes was found in *Opn4^WT* tumor-bearing mice compared to the remaining groups (Fig. S1a–c). The absolute number of monocytes did not differ between the groups, but increased frequency of monocytes was found in *Opn4^WT* tumor-bearing mice compared to sham controls (Fig. S1 d–e). The absolute number of granulocytes was higher in *Opn4^WT* tumor-bearing mice compared to sham animals. Granulocyte frequency was also higher in *Opn4^WT* inoculated mice compared to the remaining groups (sham control and *Opn4^KO* inoculated mice, Fig. S1 f, g). Remarkably, the levels of red blood cells, hemoglobin, and platelets were severely reduced in *Opn4^WT* inoculated mice in comparison to the remaining groups (Fig. S1 h–k).

Flow cytometry was used to assess the population and subpopulations of tumor-associated immune cells in the tumor microenvironment (TME) of both genotypes (Fig. S2). Macrophages are known to be highly specialized in the removal of debris and cells and to present antigens, thus playing an important role in the initial immune response[30]. No difference in total tumor-associated macrophages, M1, and M2, frequencies between the tumors were found (Fig. 1d–f). Tumor-infiltrating lymphocytes play an important role in tumor development, as in the first stages of tumor growth the immune system can combat and kill tumor cells, thus halting tumor growth. However, as the tumor progresses, tumor-infiltrating lymphocytes are often inactivated by the TME, thus leading to the exhaustion of tumor-infiltrating lymphocytes, and consequently accelerated tumor growth[31,32].

Increased frequency of CD4+ and CD8+ in *Opn4^KO* tumors compared to *Opn4^WT* was found, which is suggestive of increased migration of lymphocytes to the tumor site (Fig. 1g–n). Compared to the naive population, memory cells are known to be more persistent. While central and effector memory cells often display several phenotypic similarities, central memory cells persist longer than effector cells, which undergo a significant decrease in terms of population but display immediate cytotoxic effector function. On the other hand, central memory cells retain proliferative ability with little effector capacity, upon a reencounter with an antigen[33,34]. No difference between CD4+ naive and central memory lymphocytes was detected whereas an increase in the frequency of CD4+ effector memory cells was found in *Opn4^KO* tumors (Fig. 1h–j). We also found an increased frequency of naive, central, and effector memory CD8+ T lymphocytes in *Opn4^KO* tumors compared to *Opn4^WT* ones (Fig. 1 k–n). Intriguingly, decreased gene expression of both pro- (*Il-1β* and *Il-6*) and anti-inflammatory (*Il-10* and *Tgf-β*) players, as well as T CD8+ dependent effector function genes such as granzyme (*Gzma*) and perforin (*Prf1*) in TME of *Opn4^KO* was found when compared to *Opn4^WT* tumor-bearing mice (Fig. 1 o–t). These data suggest that as tumor growth is less marked in

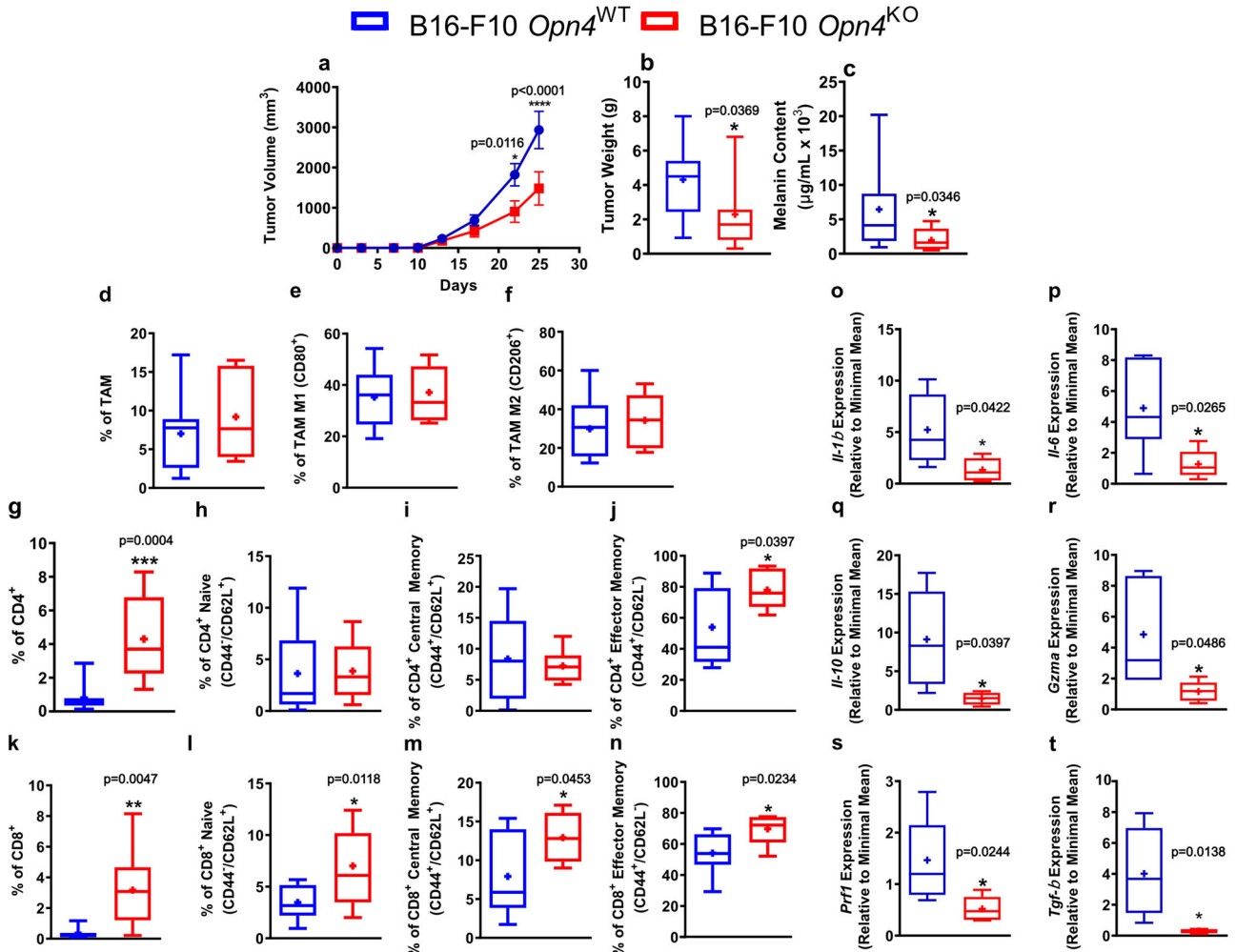

**Fig. 1 In vivo tumor growth is reduced in mice inoculated with *Opn4*KO cells and correlates with higher immune system response. a–c** Tumor volume, weight, and melanin content in *Opn4*WT and *Opn4*KO tumor-bearing mice. In **a**, $n = 17$ and 11 for *Opn4*WT and *Opn4*KO tumors, respectively. Error bars are shown as SEM; in **b**, $n = 13$ and 7, respectively; in **c**, $n = 11$ and 7, respectively. **d–n** Assessment of tumor microenvironment (TME) population represented by tumor-associated macrophages (TAM), CD4+ T lymphocytes, and CD8+ T lymphocytes. Subtypes of each cell population are indicated in the Y axis. Representative gate strategy is shown in Figure S2. In **d–f**, $n = 13$ and 7 for *Opn4*WT and *Opn4*KO groups, respectively; (**g**, **h**), $n = 12$ and 6, respectively; in **i–j**, $n = 13$ and 6, respectively; (**k–l**), $n = 12$ and 7, respectively; (**m**), $n = 11$ and 6, respectively; (**n**), $n = 13$ and 5, respectively. **o–t** Gene expression of pro- and anti-inflammatory markers as well as T CD8+ dependent effector function in TME of *Opn4*KO versus *Opn4*WT tumors. In **o**, $n = 5$ for *Opn4*WT and *Opn4*KO tumors; in **p**, $n = 6$ and 5, respectively; in **q**, $n = 6$ and 4, respectively; in (**r**), $n = 5$; in **s**, $n = 7$ and 5, respectively; in **t**, $n = 8$ and 5, respectively. In every analyzes, the n number is derived from independent samples. Asterisks represent significant differences between *Opn4*KO and *Opn4*WT tumors.

*Opn4*KO cells, one might speculate a better resolution of the inflammatory process, thus leading to a less inflamed TME.

Immune system evaluation was also performed in spleens of tumor-bearing and sham control mice. No difference in the frequency of macrophages between *Opn4*WT and *Opn4*KO spleens as well as compared to sham control mice was found. However, an increase of M1, but not M2, macrophage frequency in spleens of both genotypes compared to sham control mice was found (Fig. S3 a–c). With regards to CD4+ lymphocytes, a frequency reduction in *Opn4*WT tumor-bearing mice compared to *Opn4*KO and sham control mice was found. A reduction in the frequency of naive CD4+ was accompanied by increased CD4+ central memory population in spleens of both tumor-bearing mice compared to sham control mice; however, spleens of *Opn4*KO tumor-bearing mice showed a higher frequency of CD4+ effector memory compared to the remaining groups (Fig. S3 d–g). A similar profile of T CD8+ cells was also found as a reduction of CD8+ total frequency and naive cells, followed by increased

central and effector memory cell frequency, was seen in spleens of both tumor-bearing mice compared to control sham animals (Fig. S3 h–k).

Taken altogether, we found marked differences in the immune profile between the tumor genotypes. *Opn4*KO bearing mice showed increased CD4+ and CD8+ infiltration to the TME. Moreover, increased CD4+ effector memory presence was found in spleens of *Opn4*KO bearing mice. Collectively, these data suggest an increased immune system activity in the TME that may contribute to the reduced tumor growth found in *Opn4*KO tumors.

**Removal of OPN4 reduces metabolic activity, delays proliferation and cell cycle progression, and impairs the molecular clock in vitro.** Interested in comprehending the reduced growth of *Opn4*KO tumors, we focused on in vitro experiments. Metabolic activity and proliferation of *Opn4*KO melanocytes were significantly lower compared to *Opn4*WT counterparts (Fig. 2a).

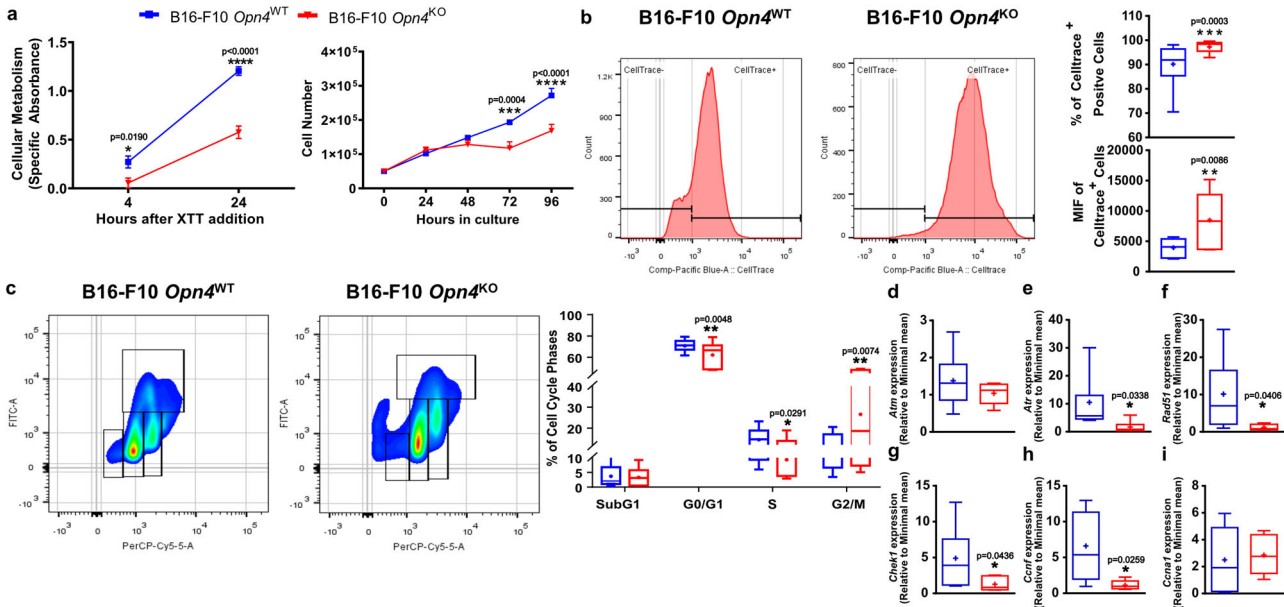

**Fig. 2 In vitro growth of *Opn4*^KO malignant melanocytes is reduced and is associated with delayed cell cycle progression. a** Metabolic evaluation using daily XTT assay for 4 and 24 h ($n = 12$ for each group) and proliferation assay by cell counting for 96 h ($n = 6$ for each group). **b** Proliferation assay by Celltrace™ staining using flow cytometry. Celltrace™ positive cells are represented. In **a** and **b**, error bars are shown as SEM. Quantitative analysis of Celltrace™ positive cells and median intensity fluorescence (MIF) of *Opn4*^WT and *Opn4*^KO malignant melanocytes ($n = 15$ for each group). **c** Evaluation of cell cycle populations and expression of cell cycle-related genes in *Opn4*^WT and *Opn4*^KO malignant melanocytes. Negative control is shown in Fig. S2. Arrows show the respective cell cycle phase. Boxplots show the quantitative evaluation of cell cycle phases by flow cytometry ($n = 18$ and 15 for *Opn4*^WT and *Opn4*^KO cells, respectively). **d–i** Expression of cell cycle-related genes in *Opn4*^WT and *Opn4*^KO malignant melanocytes. In **d**, **e**, $n = 11$ and 6 for *Opn4*^WT and *Opn4*^KO, respectively; in **f**, $n = 11$ and 5, respectively; in **g**, $n = 11$ and 6, respectively; in **h**, $n = 9$ and 5, respectively; in (**i**), $n = 10$ and 6, respectively. In every analyzes, the n number is derived from independent samples.

To quantify the proliferative capacity, cells were loaded with CellTrace™, a proliferative cell marker. As cells proliferate and divide, a reduction of the median intensity of fluorescence (MIF) is expected while slower proliferative cells show increased MIF. Corroborating our previous data, *Opn4*^KO malignant melanocytes displayed an increased percentage of CellTrace™ positive cells and fluorescence compared to *Opn4*^WT cells (Fig. 2b). Cell cycle evaluation was also performed using the 7-AAD (DNA marker) and BrdU (S Phase marker) dual staining procedure. *Opn4*^KO malignant melanocytes displayed reduced G0-G1 and S phase cell populations and increased G2/M population compared to *Opn4*^WT counterparts (Fig. 2c).

Based on our previous study[27], 6 genes were selected for qPCR validation. The genes ataxia-telangiectasia-mutated (*Atm*) and ataxia telangiectasia and Rad3-related (*Atr*) encode proteins that act on DNA damage response and are responsible for maintaining genome integrity[35]. We found no difference in *Atm* expression while *Atr* transcripts were lower in *Opn4*^KO malignant melanocytes compared to *Opn4*^WT melanoma cells (Fig. 2d, e). Rad51 is a protein that plays a major role in homologous DNA recombination during a double-strand break[36]. We found a reduction of *Rad51* in melanoma *Opn4*^KO cells (Fig. 2f). Checkpoint kinase 1 (*Chek1*) gene encodes a serine/threonine-specific protein kinase that is involved in DNA damage response and may elicit cell cycle arrest, DNA repair, and death[37]. Cyclin F, encoded by *Ccnf*, has an important role in the cell cycle as this cyclin binds and activates cyclin-dependent kinase, thus leading to cell cycle progression[38,39]. *Ccna1* (Cyclin A1) encodes a protein that is responsible for activating cyclin-dependent kinases, thus playing a positive role in the cell cycle[40,41]. In our model, *Chek1* and *Ccnf* expression was reduced while *Ccna1* expression was not affected in *Opn4*^KO cells compared to *Opn4*^WT malignant melanocytes (Fig. 2g–i).

Clock genes have been previously implicated in the melanoma carcinogenic process with exciting results[16,18,20,21]. Increased *Bmal1* gene expression, frequency, and fluorescence (MIF) of BMAL1 positive cells were found in *Opn4*^KO malignant cells compared to *Opn4*^WT cells (Fig. 3a, b). Interestingly, the microphthalmia-associated transcription factor (MITF) – the master regulator of several biological processes of melanocytes[42] – gene and protein expression, but not the frequency of positive cells, were severely less expressed in *Opn4*^KO melanocytes (Fig. 3c, d).

As important clock genes showed decreased expression, the investigation of the functionality of this system was performed. Since B16-F10 cells show a reduced synchronization capacity to different methods[18,19,43], we focused on the evaluation of the acute clock response in this assay. *Opn4*^WT and *Opn4*^KO cells were challenged with dexamethasone and forskolin, both known modulators of the molecular clock[44], using a *Per1*:Luc bioluminescence reporter assay, as previously described[25,27]. As expected, dexamethasone increased cellular metabolism and *Per1* bioluminescence in *Opn4*^WT melanocytes compared to control *Opn4*^WT cells. Forskolin, on the other hand, led to a reduction in cellular metabolism and bioluminescence in *Opn4*^WT melanocytes compared to control *Opn4*^WT cells (Fig. 3 e–h). One would expect an increase in *Per1* bioluminescence in response to forskolin, as previously shown for murine melanocytes[27]. However, the lack of *Per1* bioluminescence induction may be related to an already upregulated cAMP signaling, which is a common feature known in melanoma cells[45]. Interestingly, *Opn4*^KO malignant cells were less responsive to both treatments (Fig. 3 e–h). These data suggest that the acute molecular clock response is impaired in the absence of *Opn4*. Further circadian-related experiments are needed to clarify the impact of OPN4 in the regulation of the circadian clock function.

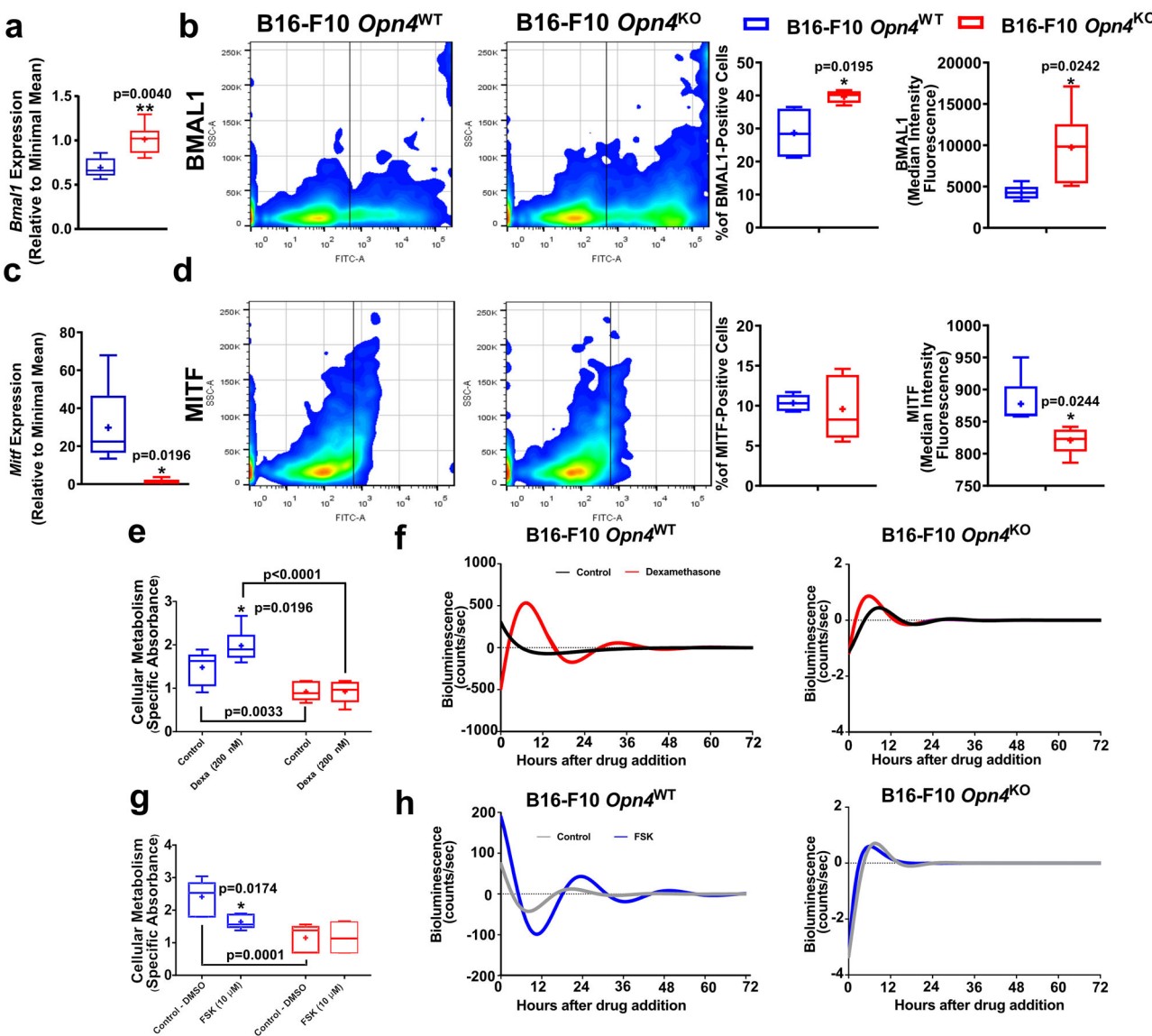

**Fig. 3 Reduced growth of *Opn4*^KO cells in vitro is associated with reduced MITF expression and reduced clock gene activation by classical activators.**
**a**–**d** *Bmal1* and *Mitf* gene and protein evaluation in *Opn4*^WT and *Opn4*^KO malignant melanocytes in vitro. Gene expression, BMAL1 and MITF positive cells, median intensity fluorescence (MIF), and quantitative analyses are depicted. In **a**, $n = 5$ and 6 for *Opn4*^WT and *Opn4*^KO cells, respectively. In **b**, $n = 4$ and 5, respectively. In **c**, $n = 5$ and 6, respectively. In **d**, $n = 5$ for each group. **e**–**h** Cellular metabolism assessment by XTT assay in response to classical clock activators and molecular clock evaluation by bioluminescence assay (*Per1*:Luc) of *Opn4*^WT and *Opn4*^KO malignant melanocytes. In **e**, $n = 6$ and 11, respectively; in **f**, $n = 4$ and 11, respectively. In **g**, $n = 6$ for both groups. In **h**, $n = 6$ and 9, respectively. In every analyzes, the n number is derived from independent samples.

We hypothesized that the inhibition of the OPN4 signaling cascade would differentially affect cellular metabolism in both cell genotypes. Thus, to evaluate a putative signaling pathway that is triggered by OPN4 in a basal condition, classic players of OPN4 signaling pathways[46] were inhibited. Calcium and phospholipase C participation were ruled out as no difference between genotypes was found (Fig. 4 a, b). The role of cGMP, which has been previously implicated in UVA-induced pigmentary response in an OPN4-dependent manner in normal and malignant melanocytes, was evaluated[24]. CaM kinase and oxide nitric synthase (NOS) were evaluated and no difference between the genotypes was found (Fig. 4 c – d). Remarkably, upon guanylyl cyclase inhibition a reduction of cellular metabolism in *Opn4*^WT malignant melanocytes (Fig. 4e) was found while *Opn4*^KO malignant cells were insensitive to the enzyme inhibitor.

Collectively, these data suggest an impairment in the cell cycle progression and the molecular clock response associated with reduced cellular proliferation with important cell cycle regulators being affected at the mRNA level in vitro. The pharmacological approach showed that the basal cellular metabolism of malignant melanocytes is dependent on calcium, CaM kinase, NOS, and cGMP pathways. However, in the absence of OPN4, guanylyl cyclase activity seems to be impaired as no inhibitory effect was seen when the enzyme was pharmacologically inhibited. One could argue that in *Opn4*^KO cells the lack of guanylyl cyclase inhibition may be due to the already downregulated enzyme activity.

**BMAL1 and MITF gene and protein expression is affected in *Opn4*^KO tumors and favors reduced tumor growth in vivo.** As *Bmal1* has been recently shown to participate as a positive prognostic marker and a putative biomarker for immunotherapy

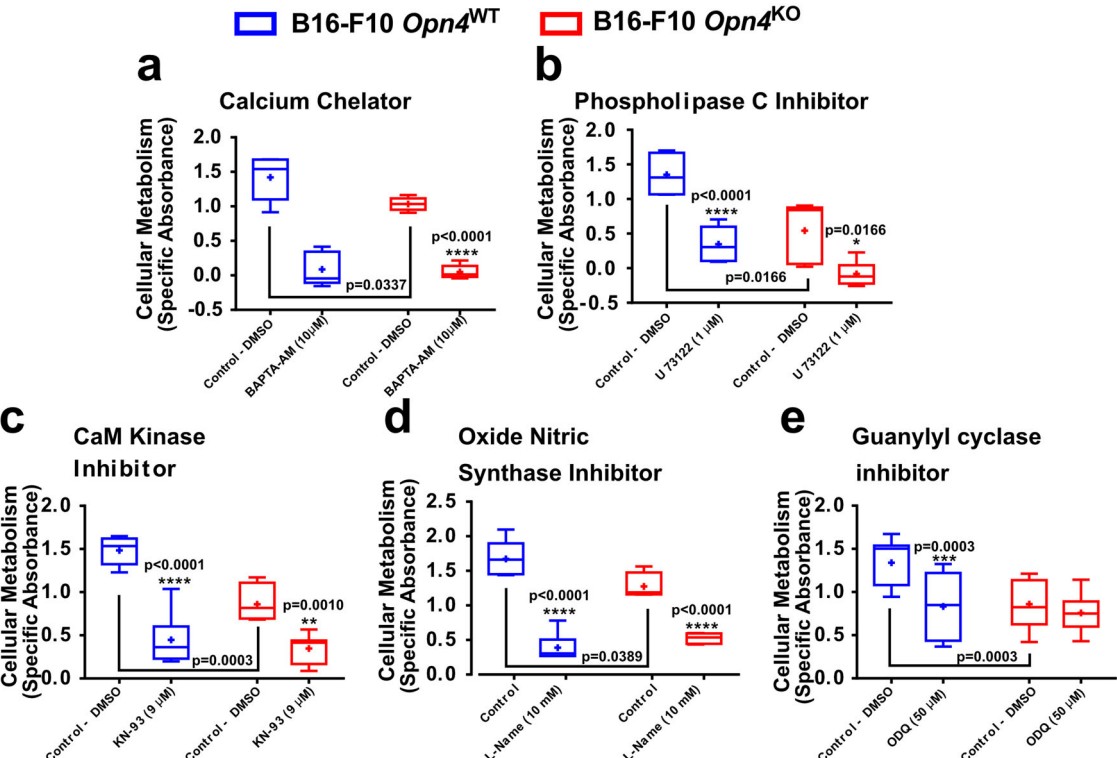

**Fig. 4 Evaluation of the OPN4 classical signaling cascade in *Opn4*WT and *Opn4*KO malignant melanocytes in vitro. a–e** Respective drugs' actions are shown in bold letters. Controls received either PBS or DMSO, depending on the drug vehicle. The highest DMSO concentration was 2%. *$p < 0.05$; **$p < 0.01$; ***$p < 0.001$; ****$p < 0.0001$. Asterisks represent differences between control and treated groups within the same genotype. Brackets represent differences between *Opn4*WT and *Opn4*KO malignant melanocytes (*p* values are shown). In **a**, $n = 6$ and 9, for *Opn4*WT and *Opn4*KO cells, respectively; In **b**, $n = 6$ and 5, respectively; in **c**, $n = 5$ and 7, respectively; in **d**, $n = 5$ and 4, respectively; in **e**, $n = 10$ and 12, respectively. In every analyzes, the n number is derived from independent samples.

success in metastatic melanoma[21], BMAL1 expression in tumor bulk and in tyrosinase-positive cells, i.e., melanoma cells, was investigated. In line with previous data, *Opn4*KO tumors displayed a higher frequency and fluorescence of BMAL1-positive cells, despite no change in *Bmal1* gene expression was detected (Fig. 5 a, b). By probing tyrosinase expression in the tumor bulk, melanoma cells (TYROSINASE-positive cells) were selected, which demonstrated higher levels of BMAL1 expression and increased frequency of BMAL1-positive cells (Fig. 5 c, d). *Opn4*KO tumors also displayed reduced *Mitf* gene expression compared to *Opn4*WT tumors (Fig. 5e), which agrees with previous experimental data (Fig. 3 a–d). Furthermore, a significant downregulation of cell cycle-related genes, such as *Atm*, *Atr*, *Ccna1*, *Chek1*, *Ccnf*, and *Rad51* was found in *Opn4*KO tumors (Fig. S4 a–f).

Despite the limitations imposed by the comparison between in vitro and in vivo data due to a myriad of different experimental conditions, conserved responses such as increased *Bmal1* gene and protein expression, and decreased *Mitf* expression in the absence of *Opn4* were observed. Indeed, reduced *Mitf* expression (gene and protein) associated with a reduction at the gene level of important cell cycle regulators suggest that *Opn4*KO malignant melanocytes continue to exhibit impaired proliferation in vivo.

We have previously shown that *OPN4* expression decreases as tumor aggressiveness increases in humans. Moreover, low *OPN4* expressing tumors also have higher levels of *BMAL1*[25], which is a biomarker for longer survival. Upon analyzing TCGA RNA-seq data[6], an association between *OPN4* and *MITF* was found. Low expressing *MITF* melanomas, which are mostly metastatic, had a significantly decreased expression of *OPN4*. A difference in *MITF* expression between human tumor types, i.e., primary and metastatic melanomas, was found (Fig. S5 a, b), which is

expected[42]. Using cell cycle-specific gene expression signature, we inferred the abundance of G1-S and G2-M related genes in each sample and showed that tumors with low *MITF* (low *OPN4*) expression display a reduced G1-S / G2-M ratio, indicating a higher proportion of cells in G2-M (Fig. S5c). We used the CIBERSORT deconvolution algorithm to estimate the abundance of different immune cell types in each sample using the RNA-seq data[47]. Low *MITF* expressing tumors displayed increased infiltration of CD4+ T memory cells, and M2 macrophages while high *MITF* tumors display higher infiltration of T gamma cells, natural killer, and mast cells frequency (Fig. S5d). Conversely, a negative correlation between *OPN4* levels and the abundance of several immune cells, such as B and CD4+ lymphocytes, and M1 macrophages was found (Fig. S5e); a positive correlation was found for mast cells and neutrophils in human melanoma (Fig. S5e).

Taken altogether, our previous studies[21,25] associated with our current data may provide a rationality why patients with low *OPN4* tumors display increased survival. In fact, reduced *OPN4* expression is associated with reduced *MITF* and increased *BMAL1* expression. In our study, decreased *MITF* and *OPN4* (also higher *BMAL1*) gene expression is associated with a slower proliferation, higher antitumorigenic TME, and therefore, reduced tumor growth.

**Protein set enrichment analyses of *Opn4*WT and *Opn4*KO proteomes reveal specific alterations that contribute to reduced growth in *Opn4*KO tumors.** To elucidate the dynamic molecular changes and screen for molecular signatures in the in vivo *Opn4*WT and *Opn4*KO tumors, the proteomics approach was

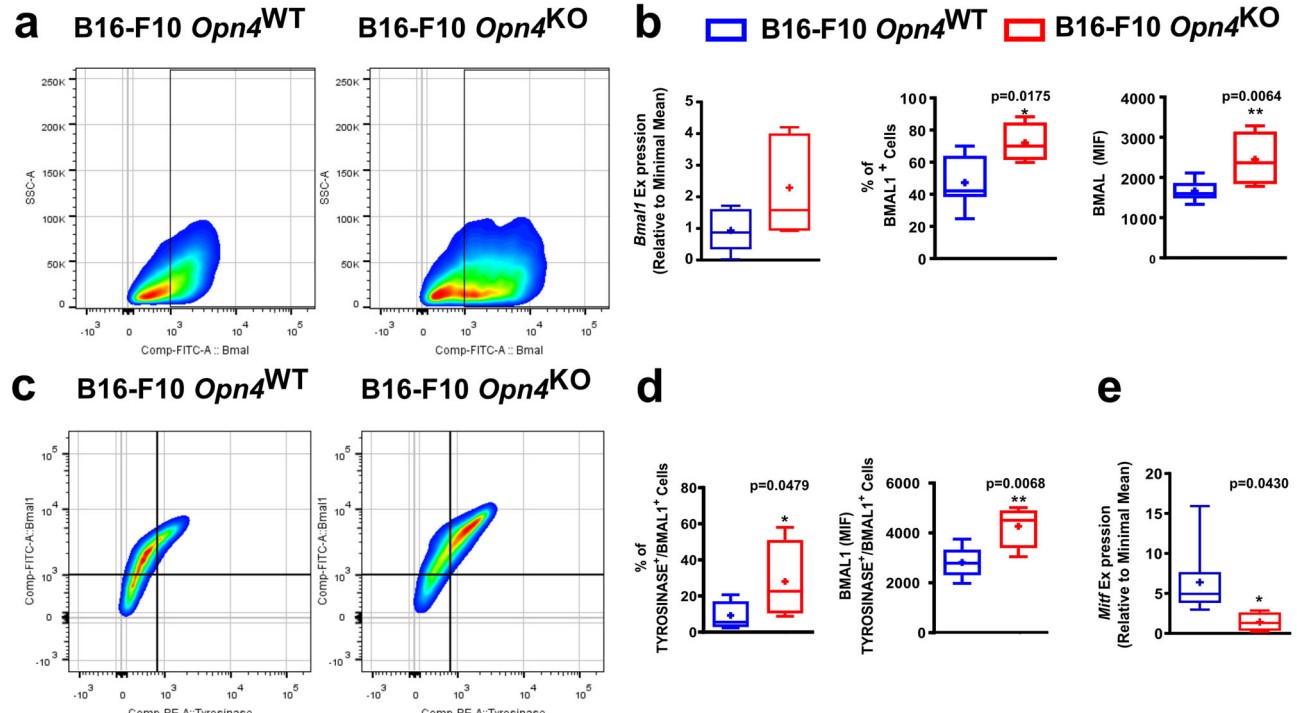

**Fig. 5 Evaluation of BMAL1 in tumor bulk and in TYROSINASE-positive cells in *Opn4*<sup>WT</sup> and *Opn4*<sup>KO</sup> tumors.** BMAL1 and MITF gene and protein expression. **a** and **c** Representative gates of flow cytometry assay. **b** Gene expression of *Bmal1* and frequency and fluorescence of BMAL1 positive cells. **d** Percentage and quantification of BMAL1 fluorescence in TYROSINASE and BMAL1 positive cells in *Opn4*<sup>WT</sup> and *Opn4*<sup>KO</sup> tumors. **e** Gene expression of *Mitf* in tumor bulk. In **b** (qPCR), *n* = 8 and 5, for *Opn4*<sup>WT</sup> and *Opn4*<sup>KO</sup> tumors, respectively. In **b** and **d** (FACS), *n* = 8 and 4, respectively; (**e**), *n* = 8 and 4, respectively. In every analyzes, the n number is derived from independent samples.

performed using differentially encoded protein analysis within label-free quantification (Fig. 6a). In total, 1480 proteins were identified in both tumors, of which 275 and 213 were unique to *Opn4*[WT] and *Opn4*[KO] tumors, respectively. We also identified 992 proteins that were shared by *Opn4*[WT] and *Opn4*[KO] tumors. Protein set enrichment analyses showed that these shared proteins belong to several biological processes such as RNA binding, GTP binding, tricarboxylic acid cycle, translation initiation, and ATP metabolic processes (Fig. 6b; Supplementary Data 1 and 2).

Protein set enrichment analyses of the exclusive proteins in *Opn4*[WT] tumors suggest processes associated with higher translation, proliferation, and aggressiveness (Fig. 6 c). The most enriched GO-term in *Opn4*[WT] was "formation of translation preinitiation complex" represented by eukaryotic translation initiation factor 3 subunits (Fig. 6 d). Indeed, the overexpression of eukaryotic translation initiation factor has been associated with hyperactivation of the translation initiation machinery[48], and therefore it may be associated with the rapid proliferation of *Opn4*[WT] tumors. Enrichment of GTPase-mediated signal transduction process in *Opn4*[WT] tumors indicates faster melanoma progression as overexpression of small GTPases has been associated with melanoma growth, aggressiveness, and diminished response to cancer drugs[49]. This protein class acts as molecular switches that cycle between active GTP-bound and inactive GDP-bound forms, interacting with downstream effectors to trigger signaling pathways[50]. For instance, overexpression of the small GTPase RHOC has been reported to accelerate melanoma progression via mechanisms that regulate PI3K/AKT and ROCK signaling pathways[51].

In addition, biological processes related to cell survival such as "telomere maintenance", "DNA replication", and "double-strand break repair" were also identified in *Opn4*[WT] tumors. For instance, poly-ADP-ribosyltransferase (PARP1) promotes

poly-ADP ribosylation of target proteins that participate in DNA repair and chromatin remodeling[52,53]. Indeed, the role of PARP1 in melanoma is dependent on MITF signaling[54]. PARP1 and OTU domain-containing protein 5 (OTUD5, called also DUBA) were shown to suppress IL-17 synthesis in CD4+ T cells (Th17)[55]. These proteins were grouped as "negative regulation of interleukin-17 secretion" in *Opn4*[WT] tumors. On the other hand, increased levels of *Tgf-ß* and *Il-6* in *Opn4*[WT] tumors (Fig. 1) could result in CD4+ naive differentiation into Th17 cells[56,57]. One might consider that increased Th17 cells would lead to enhanced tumor growth as supported by experimental evidence. However, in fact, anti-tumoral effects of Th17 in melanoma have also been described[58]. But since the CD4+ Th17 population was not evaluated in this study, further investigation is required.

On the other hand, the proteome signature of exclusive proteins from *Opn4*[KO] tumor points to anti-tumoral effects. In *Opn4*[KO] tumors "positive regulation of axon extension" and "myelin assembly" processes were identified (Fig. 6d). One can suggest that such processes may result in an attempt of tumor cells to improve tumor innervation. Axon outgrowth in neurons is mediated by TRPV2 channel[59], a protein identified in *Opn4*[KO] tumors (Fig. 6c). By contrast, in human melanoma there is a significant reduction of intratumoral nerve fibers[60]. TRPV2 activation may also inhibit, in a $Ca^{2+}$-dependent manner, small GTPase activation (e.g., RAC1)[61]. Of note, enrichment of "cellular response calcium ion" suggests an upregulation of the $Ca^{2+}$ signaling pathway in the absence of *Opn4*.

Intriguingly, enrichment of the "activation of GTPase activity" process (TIAM1, AKT2, NDEL1, and CORO1C) was also identified in *Opn4*[KO] tumors. Such proteins are known to increase activation of small GTPases (GTP-bound form), like RAC1, CDC42, and RALA[62–65]. We suggest that the proteins involved in the "activation of GTPase activity" may represent a

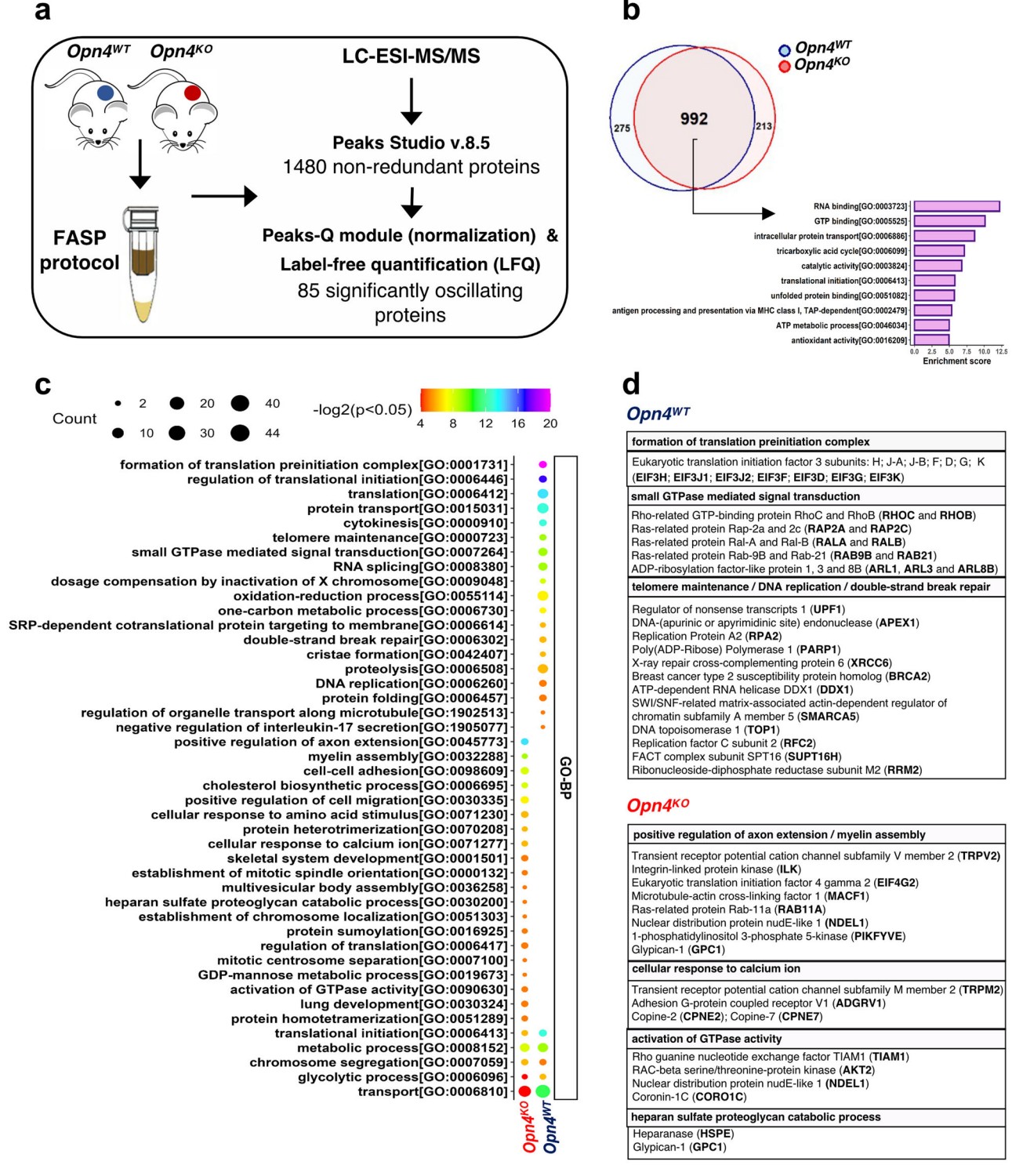

**Fig. 6 Overview of proteomic experimental design and selected results. a** Experimental protocol. **b** Venn diagram representing the overlap of results revealed by proteomes of *Opn4*^WT and *Opn4*^KO tumors and top 10 functional annotation clustering enriched (Classification Stringency: low) of commonly identified proteins in the tumor tissues performed in DAVID ($p < 0.05$). **c** Enrichment analyses of differentially identified proteins in the tumor proteomes using Gene Ontology (biological process, GO-BP) s performed in DAVID ($p < 0.05$). GO-BP terms were filtered for redundancy using REVIGO. **d** Detailed view of proteins related to some enriched terms (GO-BP) showing distinct molecular features between *Opn4*^WT and *Opn4*^KO tumor are shown. $n = 4$ for each group comprised of independent samples.

compensatory mechanism of reduced small GTPase levels in *Opn4*^KO tumors, as will be seen below. However, this event may contribute to the antitumoral effects seen in *Opn4*^KO tumors. In fact, it has been shown that TIAM1 and CORO1C are negative regulators of metastatic melanoma growth[66,67].

Our next step was to apply the label-free quantification method to relatively quantify the commonly identified proteins between tumors. A total of 85 proteins out of 992 common proteins were significantly different, being 59 and 26 proteins classified as up- and downregulated in *Opn4*^WT and *Opn4*^KO tumors, respectively

(Fig. 7 a; Supplementary Data 3). For instance, among the upregulated proteins in $Opn4^{WT}$ tumors, GTPases (e.g. RAC1, RAB5C, and SAR1B) [68,69] and proteins involved in DNA replication and cell proliferation (MCM3 and MCM5) were identified[70]. On the other hand, SOD1 protein, known to inhibit RAC1[71], was upregulated in $Opn4^{KO}$ compared to $Opn4^{WT}$ tumors (Fig. 7 b).

Label-free quantification data show differences between the proteome of both tumors and suggest a remarkable effect on GTPase activity. Furthermore, increased levels of Complement factor B in $Opn4^{WT}$ tumors compared to $Opn4^{KO}$ along with exclusively C5 and C4b proteins in $Opn4^{WT}$ proteome were identified. It is known that complement system activation in cancer is linked with a higher inflammatory TME and melanoma progression[72,73]. Therefore, these data together suggest a higher inflammatory TME, which could explain the higher tumor growth of $Opn4^{WT}$ tumors.

We also evaluated the functionality of proteins unique to each tumor genotype and those present in both proteomes but differentially regulated, using some GO-categories keywords (e.g., "cell cycle", "rhythm", "circadian", and "melanin") to identify associated proteins, following manual curation (Supplementary Data 4). Most proteins of the upregulated or exclusively identified in $Opn4^{WT}$ tumor were associated with positive regulation of G1/S progression (APEX1, GSPT2, PES1, RALA, RALB, RHOB, TBCD, UPF1, USP9X, and RPL17) and negative regulation of G2/M transition (PSMC6). On the other hand, differentially expressed proteins found in $Opn4^{KO}$ tumors were associated with positive regulation of G2/M transition (EGFR, MKI67, RAB11A, and VPS4B) and negative regulation of G1/S progression (GPNMB and EIF4E) (Fig. 7c; Supplementary Data 4). These data, therefore, corroborate the slower cell cycle progression of $Opn4^{KO}$ cells.

Although main clock proteins were not identified, several players (DBP, NDUFA9, PTGDS, TOP1, TOP2a, USP9x, and PP1CB) directly and indirectly involved in the positive loop regulation of the molecular clock (BMAL/CLOCK) were upregulated in $Opn4^{WT}$ (Supplementary Data 4). In the absence of OPN4, increased protein expression of HNRNPD, a negative regulator of the molecular clock, was identified (Supplementary Data 4). Moreover, well-known target proteins of MITF in melanogenesis such as TYRP1, DCT, and TYR1 proteins were positivity regulated in $Opn4^{WT}$ tumor (Fig. 7d; Supplementary Data 4), and therefore, strongly suggest a reduction of MITF signaling in $Opn4^{KO}$ tumor.

Taken altogether, proteomic data brought to light possible mechanisms of slower proliferation and cell cycle progression in $Opn4^{KO}$ tumors. Moreover, proteomic data suggest a highly immune suppressor TME in $Opn4^{WT}$. This event associated with a faster proliferation explains the faster growth of $Opn4^{WT}$ tumors. All these findings further provided robust evidence of an intriguing role of OPN4 in a light- and thermo-independent fashion that can be appreciated as an oncogene in melanoma.

## Discussion

We provided evidence that a light- and thermo-sensing protein, OPN4, whose role as an important light sensor responsible for circadian entrainment has been well established[74,75], plays a pro-tumoral role in melanoma. Extra-retinal OPN4 has been described in skin cells, blood vessels, and other peripheral tissues (reviewed in[22]). Some studies have shown that opsins, mainly encephalopsin (OPN3), also exert regulatory functions in cellular biology other than light or thermal sensors. For instance, OPN3 acts as a negative regulator of melanogenesis through melano-cortin 1 receptor (MC1R) interaction[76] and participates in

apoptotic processes of human melanocytes[77]. Contributing to the new role of opsins, our group has recently demonstrated that OPN4 removal in malignant melanocytes resulted in insensitivity to UVA-induced effects such as melanin content increase and apoptosis[25]. In normal melanocytes, OPN4 removal resulted in the loss of UVA-induced reduced cellular growth and pigmentation [25] and a higher proliferation and faster cell cycle progression, which was associated with higher $Mitf$ expression[27].

Our results show that the removal of OPN4 in mouse melanoma results in reduced proliferation and cell cycle progression impairment, which differs from the previous results in normal melanocytes[27]. In human melanoma, $OPN4$ expression decreases with disease progression, and tumors expressing low levels of OPN4 also display increased levels of $BMAL1$[25]. Patients with high $BMAL1$ expressing tumors showed increased survival compared to $BMAL1$ low expressing ones, a phenotype associated with increased mutational load[21]. Consequently, higher $BMAL1$ expressing tumors are more immunogenic, which affects patient survival[21]. Remarkably, the $BMAL1$ gene was also shown to be a biomarker of immunotherapy success in metastatic melanoma[21].

Furthermore, previous results from our group showed that the removal of $Trpa1$ channel results in reduced tumor growth via increased CD8+ cytotoxicity[78]. Collectively, we suggest that pharmacological modulation of either $Opn4$ or $Trpa1$, as well as others opsin-associated signaling pathway players[79], may become interesting pharmacological targets in melanoma treatment.

In vivo tumor growth corroborated our in vitro findings. Reduced tumor growth was followed by decreased melanin content, which is also in line with the literature[80,81]. Despite the increased frequency of tumor-infiltrating lymphocytes in TME of $Opn4^{KO}$ tumors, gene expression of tumor bulk suggests a less inflammatory TME, a fact that can be associated with a higher activity of the immune system and success of the immune system against tumor cells. However, we did not investigate the mechanisms underlying the slower growth of $Opn4^{KO}$ tumors. We also found evidence that the molecular clock acute response is impaired in in vivo tumors, which is in line with the in vitro data. In accordance with our previous data, we found that $Opn4^{KO}$ tumor bulk cells display a higher frequency of BMAL1 positive cells with increased protein expression; moreover, in TME we also found that tyrosinase and BMAL1 positive cells (malignant melanocytes) are also enriched in $Opn4^{KO}$ tumors compared to $Opn4^{WT}$ ones. Importantly, $Opn4^{KO}$ tumors also show impaired cell cycle-related gene expression, which argues for a slower cell cycle progression in vivo as found for the malignant melanocytes in vitro.

The in vitro data demonstrated that reduced cell proliferation is associated with reduced cell metabolism and proliferative capacity. Of note, cell cycle progression was also impaired in the absence of $Opn4$, as these cells exhibited a decrease in G1 and S as well as an increase in the G2/M phases. These increased levels of G2/M could in fact represent a point of cell cycle arrest, related to the absence of $Opn4^{KO}$. Therefore, our findings suggest that $Opn4^{KO}$ malignant melanocytes display an impaired capacity to fully start a new cell cycle. Important cell cycle regulators were also affected at the mRNA level, which collectively confirm a slower proliferation phenotype.

Gene and protein levels of molecular clock components were also differentially expressed in the absence of $Opn4$. Moreover, the molecular clock of $Opn4^{KO}$ malignant cells was less sensitive to dexamethasone and forskolin. The experimental design chosen in this study focused on the evaluation of the acute response of the molecular clock and not on the rhythmicity aspect in the absence of $Opn4$ since the B16-F10 circadian clock has previously shown weak responses to synchronizing agents[18,19]. Therefore, our experimental data are insufficient to evaluate the alterations

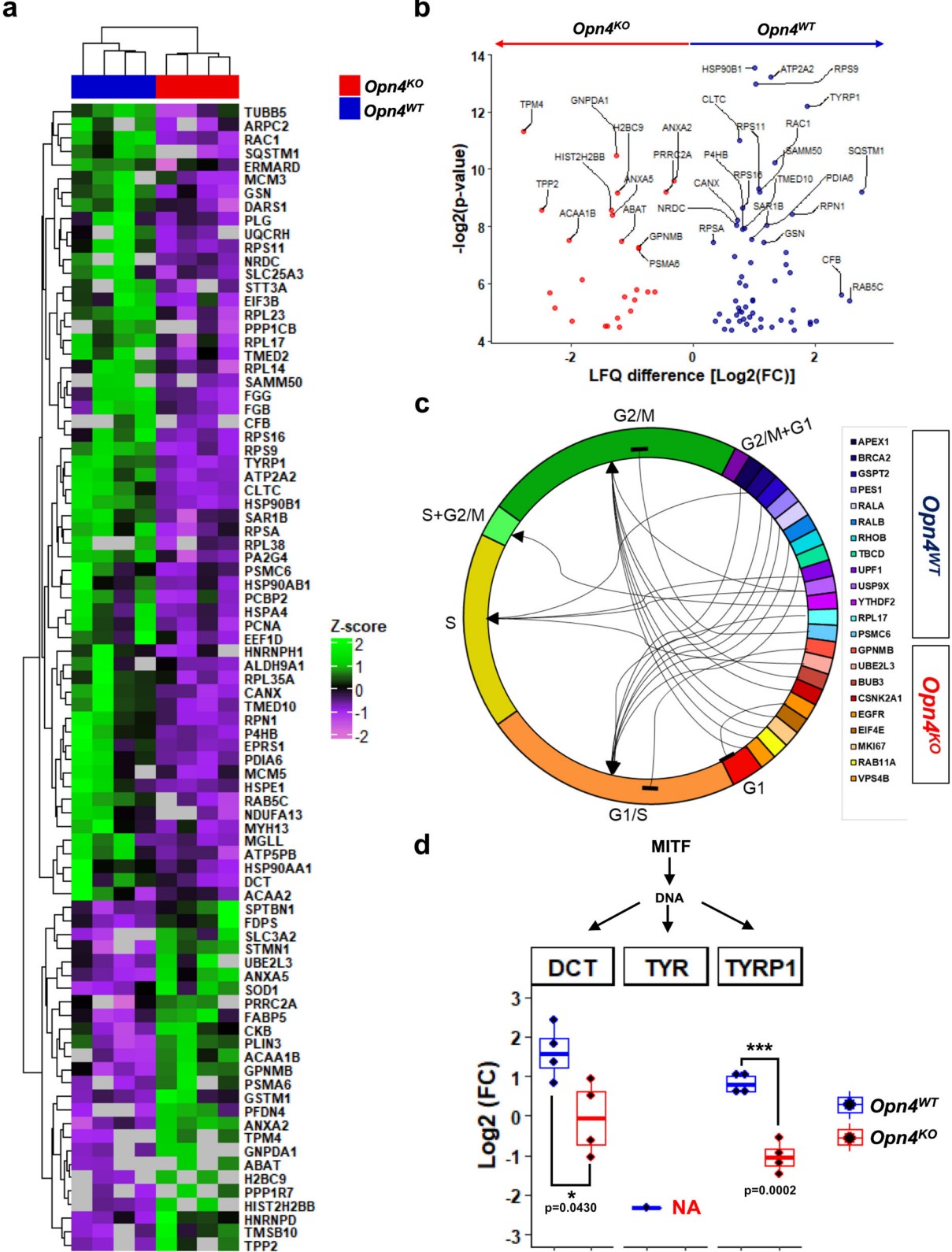

**Fig. 7 Identification of differentially expressed proteins and their participation in cell cycle regulation and MITF signaling. a** Heatmap of 85 differentially regulated proteins in $Opn4^{WT}$ and $Opn4^{KO}$ tumors. Significance was determined by two-sided t-test ($p < 0.05$) and FC ≥ 1.2 [log2(FC) ≥ 0.26 or ≤ −0.26]. **b** Scatter plot showing the distribution of significantly regulated proteins in LFQ analysis. Proteins with higher fold change and statically significant values are shown. **c** Differentially encoded proteins among melanoma types showed GO-term related to different cell cycle phases. This shows the specific relationship between proteins and cell cycle progression. Arrows and blunt arrows represent positive and negative regulation, respectively. **d** Boxplots of MITF target proteins (DCT, TYR, and TYRP1) differentially encoded by $Opn4$ absence, which were associated with melanogenesis (tyrosine metabolism) pathway and melanoma. $n = 4$ for each group comprised of independent samples.

that the absence of *Opn4* may cause to the circadian clock function. Noteworthy, the *Bmal1* gene and protein expression was increased in *Opn4*[KO] compared to *Opn4*[WT] counterparts. One can further suggest that increased acute clock gene response may represent an advantage for tumor growth. Therefore, our data show that the proliferative capacity of malignant melanocytes seems to be dependent on *Opn4* as in the absence of this protein, proliferation and cell metabolism were significantly lower.

We further suggest that the impairment of guanylyl cyclase activity demonstrated in vitro is associated with the decreased proliferation of *Opn4*[KO] cells. Proteomics analyses further corroborated these findings. In *Opn4*[KO] tumors, the RAC1 level was reduced compared to *Opn4*[WT] tumors. RAC1 is known to activate guanylyl cyclase and increase cGMP via RAC/PAK/GC/cGMP pathway[82]. Furthermore, enrichment of calcium-dependent processes suggests a higher concentration of this ion in *Opn4*[KO] tumors, and consequently, increased inhibition of guanylyl cyclase activity[83]. Experimental data strongly suggest that decreased guanylyl cyclase-cGMP activity is associated with reduced tumor growth and proliferation[84,85]. Within this line, our in vitro pharmacological guanylyl cyclase inhibition was effective only in the presence of OPN4, thus suggesting that this enzyme activity in *Opn4*[KO] is at an already low level. Therefore, different layers of evidence suggest impaired guanylyl cyclase activity in the absence of OPN4, which we suggest being one of the mechanisms by which removal of OPN4 reduces tumor growth and proliferation.

Our data provided an interaction between OPN4 and MITF, which was first shown in normal melanocytes[27]. Since *Mitf* expression is severely reduced in *Opn4*[KO] malignant melanocytes, and based on the literature[42], a reduction in cellular proliferation would be expected. However, the novelty of our data lies in the fact that the lack of *Opn4* resulted in reduced *Mitf* expression. In line with our results, Gaddameedhi's lab recently reported that *MITF* shows a rhythmic expression via BMAL1 interaction with the *MITF* promoter in human melanoma cells[86]. We suggest that OPN4 could participate in the circadian regulation of MITF either by interacting with BMAL1 at the DNA level and/or via downstream pathways that lead to degradation of mRNA and/or protein. Further research is needed to clarify this possible interaction.

Our experimental data well correlate with human melanoma from the TCGA database. We found that OPN4 correlates with MITF. Low *MITF* tumors display a lower ratio of G1-S/G2-M gene expression, which shows higher gene expression of G2-M related targets. This is suggestive of a slower cell cycle progression. Moreover, by estimating the frequency of immune cells in the tumor bulk, we found an interesting correlation between *MITF* and *OPN4* with different immune cell types that is suggestive of a role of OPN4 in TME and consequently tumor progression.

Proteomics analyses confirmed our experimental data at all levels and further shed light on additional pathways that collectively provide the rationality of why *Opn4*[KO] tumors show reduced proliferation. We uncovered a downregulation and upregulation of positive and negative loops of the main clock machinery, respectively, in *Opn4*[KO] compared to *Opn4*[WT] tumors. With regards to the cell cycle, proteins related to G1/S and G2/M were down- and upregulated, respectively, in the absence of *Opn4*. Evidence of a less inflammatory TME in *Opn4*[KO] tumors was found as reduced complement expression and gene expression of inflammatory cytokines were identified in *Opn4*[KO] tumors. Reduced expression of GTPases has been linked with slower cell cycle progression[87], which is in line with the slower *Opn4*[KO] proliferation. Moreover, the higher proliferative capacity of *Opn4*[WT] tumors can be associated with the

exclusive presence of DNA repair proteins (PARP1, RPA2, APEX1, and XRCC6) and DNA replication (RCF2 and TOP1). Increased DNA repair capacity has been related with more aggressive cancer development and treatment resistance in melanoma[88–91]. We previously identified a link between higher expression of *BMAL1* and lower base excision repair score. Indeed, such association was found in our experimental model as the absence of OPN4 resulted in elevated BMAL1 protein expression as well as reduction/absence of base excision repair-related proteins (PARP1, RPA2, APEX1[92–94]; uniprot database). Since these proteins were only identified in *Opn4*[WT] tumors, we suggest a possible reduction of DNA repair via decreased base excision repair activity. It may be possible that these findings are linked with reduced MITF signaling in the absence of OPN4 via an elusive mechanism. Although MITF was not detected in the proteomics, we found that classical MITF targets such as DCT, TYR, and TYRP1 were less expressed in *Opn4*[KO] tumors.

Our findings are suggestive that OPN4 acts as an oncogene, but additional experiments using human melanoma cell lines and different approaches, i.e., knockdown, knockout, and/or rescue strategies are required to fully establish the OPN4 role as an oncogene. One must consider that our study used mice kept at thermoneutrality, and thus, comparisons with other studies that used temperature lower than the thermoneutral, should be made with caution. With regards to the circadian aspects, our in vitro and in vivo sampling collection took place within a narrow time interval, thus, ruling out the influence of time in our analyses. However, additional circadian experiments are necessary to fully establish the impact of OPN4 in the regulation of the circadian clock.

Taken altogether, our data add a novel layer of complexity to the opsin realm as we provide evidence that OPN4 can be a tumor oncogene in melanoma. The traditional concept that opsins are light sensors has been challenged as opsins can also detect thermal energy. Emerging evidence shows that opsins also display light- and thermo-independent functions likely due to protein-protein interaction, and therefore, opens a new field of investigation. Understanding the function of opsins as light- and thermo-independent proteins in both physiological and pathological contexts can yield promising therapeutic strategies and tools in a near future. Therefore, we suggest that OPN4 can be seen as a tumor oncogene in melanoma and could be pharmacologically targeted.

## Methods

**Cell culture**. Murine malignant B16-F10 *Opn4*[KO] melanocytes were generated using Clustered Regularly Interspaced Short Palindromic Repeats (CRISPR) technique. *Opn4*[WT] and *Opn4*[KO] cells were subject to *Per1:Luc* gene transfection as described previously[25] and were used in this study. Wild type B16-F10 cells were initially donated by Prof. Roger Chammas (School of Medicine, University of São Paulo). Cells were cultured in phenol red-free RPMI 1640 medium (Atena, Brazil), supplemented with 25 mM NaHCO$_3$ (Sigma-Aldrich, USA), 20 mM HEPES (Santa Cruz, USA), 10% fetal bovine serum (FBS, Atena, Brazil), 1% antibiotic/antimycotic solution (10,000 U/mL penicillin, 10,000 μg/mL streptomycin, and 25 μg/mL amphotericin B, Thermo Fisher, USA), pH 7.2. Geneticin (200 μg/mL, ThermoFisher, USA) was used to guarantee selection during maintenance and removed during the experiments. Cells were kept at 37 °C with 5% CO$_2$. In all experiments, unless otherwise mentioned, FBS was reduced to 2% and all-trans retinal (100 nM, Sigma-Aldrich, USA) was added as a supplement. For experiments in the absence of CO$_2$, HEPES concentration was increased to 50 mM. All in vitro experiments were carried out in the dark under red dim light (7 W Konex bulb and Safe-Light filter GBX-2, Kodak, USA). Unless otherwise mentioned, cells were collected on the 4th day after seeding.

**Animal handling and care**. Animal experimentation was performed according to Brazilian animal welfare regulations and approved by the Committee of Animal Ethical Experimentation of the Institute of Biosciences, University of São Paulo (Approval 322/2018). Three- to eight-months old male C57BL/6 J mice were kept under a 12:12 light/dark cycle (800–1000 lux white LED light, ranging from 420 to 750 nm) at controlled temperature (30 ± 1ºC). Lights were on at 7 a.m. and off at

7 p.m. Mice were subcutaneously inoculated in the right flank with $2 \times 10^6$ $Opn4^{WT}$ or $Opn4^{KO}$ B16-F10 cells in 100 µL of phosphate-buffered saline (PBS). Sham control animals were injected with the same volume of PBS. Then, mice were single housed for the entire experiment. Tumor measurement, food intake, and weight were assessed every three to four days at the same time of the day (from 2 to 3 pm). Length, width, and height of the tumors were measured with a caliper rule from the 13th day onwards, and the volume (mm³) was calculated according to the formula: π/6 x length x width x height[95]. Animals were $CO_2$-euthanized 25 days after inoculation, and death was assured by cervical dislocation, between 9 and 10 am (Zeitgeber time 2–3). After euthanasia, every animal was visually inspected, and no visible metastasis nodules were found. The organs and blood were harvested and immediately processed or stored at −80 ℃. The tumor was resected, weighed and melanin levels were quantified as previously described[20].

**XTT assay––metabolic evaluation**. For XTT experiments, $10^4$ $Opn4^{WT}$ or $Opn4^{KO}$ malignant melanocytes were seeded in 96-well plates in 100 µL of experimental medium and kept for 24 h in the incubator. On the following day, 50 µL of XTT and electron coupling reagent (ECR) solution (7:1) was added. For pharmacological manipulation, drugs or respective vehicles were added (Supplementary Table S1). Cells were kept at 37 ℃ with 5% $CO_2$ for 4 and 24 h, and then the solutions' absorbance was read in a spectrophotometer (SpectraMax 250, Molecular Devices, USA). Specific absorbance was calculated as $(A_{450\ sample} − A_{450\ blank}) − A_{660}$ according to the manufacturer's instructions (CyQUANT™ XTT Cell Viability Assay, Thermofisher, USA).

**Cellular growth evaluation**. Fifty thousand $Opn4^{WT}$ or $Opn4^{KO}$ malignant melanocytes were seeded in 12-well plates in the experimental medium and kept for 24 h in the incubator. Twenty-four, 48, 72, and 96 h later, the cells were harvested with Tyrode/EDTA solution and counted in a hemocytometer.

**CellTrace™ proliferative assay**. One thousand $Opn4^{WT}$ or $Opn4^{KO}$ malignant melanocytes were loaded with Celltrace dye (1 µL for every $10^6$ cells), seeded in 6-well plates in the maintenance media (10% SFB), and kept in the incubator for four days. Then, cells were harvested with Tyrode/EDTA solution, stained with Live/ Dead violet fluorescent dye™ (405 nm, 1:500, ThermoFisher, USA) in PBS and kept at 37 ℃. The cells were fixed in 4% paraformaldehyde (Electron Microscopy Science, USA) on ice for 30 min, and stored at 4 ℃ until processing in Canto II flow cytometer (BD Biosciences, USA) using DiVA 8 acquisition software. At least $10^5$ events were captured. Cells were gated using FSC and SCC, duplets were excluded using FSC-H vs FSC-A. Negative Live/Dead stained cells (viable cells) were gated and from this population, CellTrace-positive cells were calculated in terms of percentage and median intensity of fluorescence (MIF). Data were processed in FlowJo™ software (BD Biosciences, USA). Negative controls are shown in Fig. S2 K.

**Cell cycle analysis by flow cytometry**. Cell staining followed the manufacturer's instructions (BD Biosciences, USA). In brief, cells were loaded with BrdU solution (10 mM) for 2 h, and then harvested and fixed with Cytoperm Cytofix solution (BD Biosciences, USA) on ice for 30 min. Approximately $10^5$ to $5 \times 10^5$ cells/well were placed onto a 96-well round bottom plate in Cytoperm Permeabilization Buffer Plus, kept on ice for 10 min, followed by another incubation with Cytofix/Cytoperm buffer on ice for 5 min. DNAse (300 µg/mL) was added and cells were placed at 37 ℃ for 1 h. Cells were resuspended in anti-BrdU antibody in Perm/Wash buffer (FITC, 1:50) and kept at room temperature for 20 min. Cells were resuspended in 7-AAD solution and kept in staining buffer until the acquisition in Canto II Flow Cytometry (BD Biosciences, USA). In every step, cells were washed with Perm/Wash buffer, followed by centrifugation (200 x g for 3 min). Negative controls are shown in Fig. S2L.

**RNA extraction, cDNA synthesis, and quantitative PCR (qPCR)**. Total RNA was extracted from cells or tumors with Trizol (ThermoFisher, USA) according to the manufacturer's instruction, using 1-bromo-3-chloropropane (Sigma, USA), isopropanol (Sigma, USA), and washed with 75% molecular grade ethanol (Sigma, USA). DEPC water was used to resuspend the RNA pellets. Genomic contamination was removed using TURBO DNAse (ThermoFisher, USA) and RNA concentration and quality ($OD_{260}/OD_{280}$) were assessed in a spectrophotometer (NanoDrop, USA). One µg of total RNA was subject to reverse transcriptase reaction using random hexamer primers and Superscript III, in addition to the reagents recommended by the enzyme manufacturer (Life Technologies, USA).

Twenty-five ng of cDNA was subject to quantitative PCR reactions using species-specific primers (Supplementary Table S2) spanning introns, based on sequences obtained from the GenBank (http://www.ncbi.nlm.nih.gov/genbank), designed by Primer Blast (http://www.ncbi.nlm.nih.gov/genbank) or Primer Quest (IDT, USA), and synthesized by Integrated DNA Technologies (IDT, USA). *Rpl37a* was used to normalize the expression values of the genes of interest in the in vitro and in vivo assays. *Rpl37a* showed a robust and stable expression across samples (SD < 1). For TaqMan assay, reactions containing cDNA, primers, fluorescent probes, and Kapa Probe Fast Mix (Kapa Biosystems, USA) were used and run in triplicates for each cDNA sample. Reactions were carried out in an iQ5

thermocycler (BioRad Laboratories, USA) in the following conditions: 3 min at 95 ℃ followed by 45 cycles of 15 s at 95 ℃ and 60 s at 60 ℃. For SYBR Green assay, independent solutions were prepared with cDNA, specific primers, and Kapa Sybr Fast mix (Kapa Biosystems, USA), and run in duplicates in an iQ5 thermocycler in the following conditions: 10 min at 95 ℃, followed by 45 cycles of 15 s at 95 ℃, 60 s at 60 ℃, and 80 cycles of 10 s at 55 ℃ with a gradual increase of 0.5 ℃.

**Per1:Luc bioluminescence assay**. One hundred thousand $Opn4^{WT}$ or $Opn4^{KO}$ malignant melanocytes were seeded in 35 mm dishes in experimental media (containing 50 mM HEPES) and kept in a $CO_2$ incubator for 24 h. On the next day, cells were treated with dexamethasone (200 nM) or forskolin (10 µM). Drugs and vehicles remained in dishes until the end of the experiment. Luciferin (Promega, USA, 100 µM) was also added, dishes were sealed with 35 mm round coverslips (VWR, England) and parafilm, and placed into the Lumicycle equipment (Actimetris, USA) in an incubator without $CO_2$ positive pressure, at $37 \pm 0.5$ ℃. Bioluminescence was recorded every 10 min. The temperature of the incubator was monitored every 10 min (iLog, Escort Data Loggers, USA). Baseline subtracted data were plotted using a dampened sine wave function in Graphpad Prism (7.0).

**Hematological analyses**. After euthanasia, blood was collected by cardiac puncture in EDTA (10.25 mg/mL) collection tubes and immediately processed. Analyses were performed on an automated hematology analyzer (BC-2800Vet, Mindray, USA) using mouse-specific algorithms and parameters.

**Flow cytometry for tumor-associated macrophages, tumor-infiltrating lymphocytes, and BMAL positive cells**. The tumor and spleen were dissected and dissociated through a cell strainer (100 µm, Corning, USA) in PBS. Red blood cells (RBC) were lysed using ACK (ammonium-chloride-potassium) and RBC lysing buffer (0.15 M NH₄Cl, 10.0 mM KHCO₃, 0.1 mM Na₂EDTA), followed by 1000 x g centrifugation. The supernatant was removed and the remaining cells were resuspended in PBS. One million cells per well were stained in a round bottom 96 well plate using a two-step staining protocol. Flow cytometry procedures were performed as described previously[78]. In short, first, cells were stained with a Live/ Dead dye (Fixable aqua 405 nm, Invitrogen, USA) at 4 ℃ for 20 min, and after washing 100 µL of a solution containing surface antibodies diluted in staining buffer (1% FBS, 1 mM EDTA, and 0.02% NaN₃ in PBS) were added into each well. After 30 min at 4 ℃, the samples were washed (2X) and resuspended in staining buffer until acquisition. The following antibodies from Biolegend, USA (1:200 dilution) were used unless otherwise mentioned: APC-Cy7 anti-Mouse F4/80 (Cat no. 123118), PerCP-Cy5.5 anti-Mouse CD80 (Cat no. 194722), FITC anti-mouse CD206 (Cat no.141704), FITC anti-mouse CD4 (Cat no. 100509), PE-Cy7 anti-mouse CD44 (Cat no. 560569), PE anti-mouse CD62L, (Cat no. RM4304), and APC anti-mouse CD8 (Cat no. MCD0805, Invitrogen, USA). For tyrosinase and BMAL1 dual staining, tumor cells were labeled with anti-tyrosinase (1:25, goat polyclonal, Santa Cruz, USA SC 18182) and anti-BMAL1 (1:100, rabbit polyclonal, ABCAM, ab93806) in staining buffer as described above. Secondary antibodies anti-goat (555 nm) and anti-rabbit (488 nm) (both Alexa Fluor ThermoFisher, USA) were used (1:100). Samples were assessed in a FACSCanto II cell analyzer (Becton Dickinson, USA) using DiVA 8 acquisition software and FlowJo 5 V10 (Becton Dickinson, USA) data analysis software. Representative negative controls of all flow cytometry experiments can be found in Fig. S1A–I.

**Sample preparation for proteomics**. Tumor tissues were mechanically homogenized in lysis buffer (2% SDS, 100 mM Tris-HCl, pH 7.8) supplemented with protease inhibitors (Complete™ ULTRA Tablets, Mini, EASYpack Protease Inhibitor Cocktail, Roche, USA), using a POLYTRON® PT 1200 and sonicated for three cycles at 30% amplitude (20 s bursts with 20 s pauses). Samples were centrifuged at 16,000 x g for 10 min at 4 ℃ to remove the tissue debris. The supernatants were collected and the protein concentration was measured by using a BCA protein assay (Pierce™ BCA Protein Assay Kit, Thermo Scientific, USA). Aliquots of SDS-lysates containing 200 µg of total protein from each sample were processed according to the filter-aided sample preparation (FASP) method, using Microcon 10 kDa centrifugal filter units (Merck, USA) operated at 10,000 x g for 50 min at 20 ℃[96–98]. Next, Trypsin/LysC Mix (Promega, USA) was added to the filters at an enzyme-to-protein ratio of 1:100 (w/w), and incubated for 12 h at 37 ℃; a second digestion was carried out with trypsin (Promega, USA) at an enzyme-to-protein ratio of 1:100 (w/w) at 37 ℃ for 4 h. Following protein digestion, peptides were filtered through the membrane and purified with reversed-phase chromatography using C18 micro-pipette tips (TopTip™, PolyLC inc, USA), according to the manufacturer's instructions. Peptides were dried in a vacuum concentrator and stored at −20 ℃.

**Mass spectrometry data acquisition**. Dried peptides were recovered in 20 µL of 0.1% formic acid in water and analyzed on a nano-ACQUITY UPLC system (Waters, USA) coupled online to a maXis 3 G quadrupole time-of-flight (Q-TOF) mass spectrometer (Bruker Daltonics, Germany), equipped with a Captive Spray nanoelectrospray source. One microliter of the sample was injected and peptide mixture was loaded onto a trap column (nanoAcquity UPLC® 2G-V/M Trap 5 µm

Symmetry® C18, 180 µm x 20 mm) for 3 min at a flow rate of 7 µl/min of 0.1% formic acid in water. Peptides were separated on an analytical column (nanoAcquity UPLC® 1.8 µm HSS T3, 75 µm x 200 mm) using 240 min gradient from 2% to 85% of 0.1% formic acid in acetonitrile (1 min at 2%, 209 min 2-30%, 10 min 30–85%, 5 min wash at 85%, 5 min 85-2%, 10 min equilibration at 2%). The flow rate was set at 200 nL/min. Eluted peptides were analyzed in a maXis 3 G (Q-TOF) mass spectrometer operated in the positive mode under a data-dependent acquisition manner in the m/z range of 150–2200. Precursor ions were fragmented with collision-induced dissociation (CID).

**Protein identification, relative quantification, and statistical analysis**. The raw files (.d) were loaded into the Peaks Studio 8.5 software (Bioinformatics Solution Inc., Canada) and the PEAKS' standard workflow (de novo peptide sequencing, PeaksDB, PeaksPTM and SPIDER tools) was applied to database search analysis and identification of proteins[99–101]. MS/MS spectra were searched against the *Mus musculus* (mouse) reference proteome database available at UniprotKB (Proteome ID UP000000589, with 21,990 protein entries (download one protein sequence per gene, release date May 2021). The following parameters were used: precursor mass tolerance of 25 ppm; fragment mass tolerance at 0.025 Da; trypsin was set as the specific enzyme and up to two missed cleavages were required; carbamidomethylation (Cys) as fixed modification, oxidation (Met) and acetylation (protein N-terminal) set as variable modifications with maximal 3 modifications per peptide in SPIDER outcomes. Significance score of $-10\lg P > 20$ ($p$ value <0.01) for proteins and peptides and with at least one unique peptide were applied to protein identifications. Further, proteins were considered to be identified in an experimental group when they were identified in at least two out of the four biological replicates of a given group. PEAKS Q module was applied (Label-free quantification method)) to database search outcome, and the normalization factors based on Total-Ion Count (TIC) were obtained. The normalization factor obtained for each sample was applied to calculate normalized areas of proteins. Normalized protein areas have been log2-transformed values and were used to calculate Student's $t$-test, in which p-value <0.05, and Fold Change ≥1.2 [log2(FC) ≤ −0.26 or ≥ 0.26) were selected as a cut-off of significance to the differential relative abundance of proteins[102,103]. Functional enrichment analysis was performed in DAVID v.6.8 (Database for Annotation, Visualization and Integrated Discovery) using the Uniprot Entry, and then Gene Ontology-Biological Process (GO-BP) was reported ($p$ value < 0.05). REVIGO (http://revigo.irb.hr/)[104] was used to identify redundant GO terms and group the related ones together by semantic clustering to reduce redundancy, with a similarity index of 0.7 using *Mus musculus* as background.

**Bioinformatics analysis**. The Cancer Genome Atlas (TCGA) pre-processed RNA-seq and clinical data from 103 primary melanomas and 368 metastatic melanomas were downloaded from the UCSC XENA Browser[105]. RNA-seq data were generated using the Illumina HiSeq 2000 RNA sequencing platform and quantified with RSEM. Estimated counts were upper quartile normalized and $\log_2$ (normalized counts + 1) transformed. Cell cycle phase abundance scores were calculated by averaging the mean-centered expression of G1/S and G2/M genes previously described in human tumors[106]. The abundance of different tumor-infiltrated immune cells was estimated using CIBERSORTx[47] and the LM22 signature matrix[107]. CIBERSORTx was run under "Absolute mode" using batch correction ("B-mode"), 100 permutations, and quantile normalization disabled, as recommended for RNA-seq data.

**Statistics and Reproducibility**. In all analyzes, values were excluded when higher or lower than mean ± 2 standard deviation (SD). Data were checked for normality using D'agostino & Pearson omnibus normality test. When comparing two or three groups at a single timepoint unpaired two-sided Student's $t$-test or One-Way ANOVA followed by Tukey post-test, respectively, was used. For comparison between two independent variables, Two-Way ANOVA followed by Bonferroni post-test was employed. Gene expression was determined using the $2^{-\Delta\Delta CT}$ method[108]. The data were obtained from at least two independent experiments (N). Bioinformatics and proteomics statistical analyses are described above. The number of independent samples (n) is shown in each figure legend. $p$ value <0.05 were used to reject the null hypothesis and were calculated in GraphPad Prism 7.0.

All experimental data used to generate Figs. 1 to 5 and Fig. S1 to S5 can be found in the supplementary data 5. Data used to generate Figs. 6 and 7 are found in supplementary data 1 to 4. Data are graphically represented as mean ± SEM or as Box plots. For the latter, boxes extend from the 25th to the 75th percentile, the horizontal line shows the median, mean is represented by "+", and whiskers are drawn from minimum to maximum values.

**Reporting summary**. Further information on experimental design is available in the Nature Research Reporting Summary linked to this paper.

## Data availability
The raw MS data associated with this manuscript have been submitted to a public repository (the Mass Spectrometry Interactive Virtual Environment – MassIVE, http://

massive.ucsd.edu) and deposited to the ProteomeXchange Consortium (http://www.proteomexchange.org/). These data are associated with the identifier MassIVE ID MSV000088579 and Proteome Exchange ID PXD030477.

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

## Acknowledgements

This work was partially supported by the Sao Paulo Research Foundation (FAPESP, grants 2012/50214-4, 2017/24615-5, and 2018/14728-0 to Castrucci AML; 2017/05264- 7 to Câmara NOS; 2019/19435-3 to Menck CFM) and by the National Council of Technological and Scientific Development (CNPq 303078/2019-7 to Castrucci AML; 428754/2018–0 to Moraes MN). Moraes MN is a Young Investigator of FAPESP (2017/26651-9). de Assis LVM was a fellow of FAPESP (2013/24337-4 and 2018/16511-8). Lacerda JT, Domínguez-Amorocho O, Mendes D, Silva MM, and Kinker GS are fellows of FAPESP (2020/04524-8; 2017/16711-4; 2017/18781-0, 2017/24217-0, and 2019/25129-2, respectively). The authors thank Renata Alves dos Santos for providing excellent care for the mice.

## Author contributions

LVMA, MNM, and AMLC designed the study and the original hypothesis. LVMA, JTL, MNM, AOD-A, DM, and MMS acquired the experimental data. LVMA performed all in vitro experiments with the aid of DM and MMS. AOD-A performed the flow cytometry experiments with the aid of LVMA. LVMA together with GSK evaluated and analyzed the bioinformatics data. JTL and MNM processed the samples for proteomics and analyzed the data. LVMA wrote the first draft of the manuscript with the contribution of JTL and MNM. AMLC, CFMM, and NOSC critically revised the manuscript. All authors contributed to the discussion and critically revised the manuscript. AMLC supervised the study, ensuring rigorous data quality control, contributed to the discussion, and critically revised the manuscript. All authors have approved the definitive version of the manuscript and agreed to be accountable for all aspects of the study in ensuring the accuracy and integrity of any part of the study.

## Competing interests

All authors declare no competing interests.
