## [Peer Review File · Communications Biology]

Reviewers' comments:

Reviewer #1 (Remarks to the Author):

In this manuscript, the authors examine the effects of knocking out Opn4 on melanoma cell proliferation in vitro and in vivo. They find that loss of Opn4 is associated with reduced proliferation and tumor growth, which they found to be correlated with a number of changes in immune responses and Mitf signaling. Strengths of this manuscript include the relevance to melanoma (a growing and deadly problem) and the characterization of Opn4, for which there is little information currently available for its impact in cancer development. Though the authors have previously published work on Opn4 in normal melanocytes, the current manuscript is focused on metastatic melanoma cells. Knockout of Opn4 in these two model cell systems leads to different impacts on cell proliferation, which is intriguing, but not examined here.

Major issues:

1) Additional information can be provided about the generation of the Opn4 KO B16-F10 cells. For example, do the authors have any data showing the Opn4 is actually knocked out in their cell line (loss of protein by western blotting, for example)? Was a single KO clone used in all of their studies? Or, did they validate any findings on a second KO clone? Have the authors tried re-expressing Opn4 in their KO cell line to rescue the effects (and prove that the changes in the cell are due to Opn4 KO and not some other non-specific effect of using CRISPR? Any information the authors can provide will help readers better understand the authors findings and potential limitations.

Minor issues:

- 1) I would suggest some editing for English and grammar. Some examples:
a) Title: "...novel oncogene gene..." Saying 'gene' twice is not necessary
b) Line 60: Mitf, not Mift

Reviewer #3 (Remarks to the Author):

The manuscript "Melanopsin (OPN4) is a novel oncogene gene in cutaneous melanoma: evidence 2 from an experimental study" describes the role of OPN4 in the tumoral context. Authors used different in vitro and in vivo strategies in both humans and a murine model. The silencing of OPN4 reduced the tumor growth rate and the level of molecules and mechanisms involved in cell proliferation, thus, it is considered an oncogene. In addition, the work also describes the behavior of immune variables as these mechanisms regulate the outcome of this pathology. The research was very well conducted but there are some specific explanations and analysis that could improve the manuscript.

Major concerns:

Page 4: lines 130-135: Authors say: "we inoculated C57Bl/6J mice, kept in thermoneutrality ($30 \pm 1^\circ\text{C}$), with B16-F10Opn4WT or B16-F10Opn4KO cells. It has been previously shown that tumor growth is significantly faster in mice kept at temperatures below thermoneutrality than at thermoneutrality 26. Indeed, tumor growth was only measurable from the 13th day onwards unlike previously shown in B16- F10Opn4WT inoculated mice kept at 22°C 19". The habitual temperature to maintain mice is around 22°C , why do authors consider higher temperatures (30°C) normal? On the other hand, tumors tend to grow faster at higher temperatures. Authors should explain these protocols in detail in the methods section.

1) Line 174: Author said: "...decreased gene expression of both pro- (IL-1 β and Tgf- β) and anti-inflammatory (IL-6 and IL-10) players..." Tgf- β is an anti-inflammatory cytokine, and il-6, in spite of its pleiotropic functions, is classified as pro-inflammatory cytokines since it is secreted during inflammation together with il-1 β and TNF- α . Author should correct it.

Page 10, lines 351-356: Authors say: "These proteins were grouped as "negative regulation of interleukin-17 secretion" in Opn4WT tumors. Indeed, the presence of Th17 cells in TME is a

common feature in many cancers 52, 53 and the evasion of Th17 cells constitutes an immune evasion mechanism 54. Therefore, the suppressor activity of PARP1 and OTUD5 in IL-17 production in TME could be associated with the decreased TIL levels found in Opn4WT tumors (Fig. 1).” It is confusing that authors should designate a pro-tumoral (evasion prone) role to Th17 cells when these cells decrease in WT mice in which the tumors grow faster. Moreover, the role of pro-inflammatory immune mechanisms, as in those mediated by Th1, Th17 or M1 cells, in the progression/regression of tumors depends on the tumor type. A more in-depth and specific interpretation and discussion is needed about this point. Additionally, this result and the cytokine levels shown in Fig. 1 could be collectively interpreted since Tgf- β and il-6 participate in the differentiation of CD4 naive to Th17 cells (considering the differences between human and mice in this process).

Page 11, lines 385-388: Similarly, authors say: “Furthermore, significantly increased levels of Complement factor B (CFB) in Opn4WT tumors compared to Opn4KO, and with exclusively identified C5 and C4b proteins in Opn4WT proteome, could be associated with suppression of the immune response in Opn4WT melanoma by inhibiting TIL function 68.” Since the complement mechanism is also related to the inflammation process, this association should be discussed in more detail.

It is relevant to consider the complexity of the immune response in the tumor context and the particularities observed in each specific tumor. A more complete and robust discussion about the immune role is needed.

2) Since the B16-F10 tumor cell line induces pulmonary metastases and the number of tumor cells inoculated was very high (2.106 cells), authors should explain their thinking about the lack of metastasis in this case and include an explanation of the procedures by means of which they evaluated metastases. In this regards, authors say: “After euthanasia, every animal was visually inspected, and no metastasis was found”, please clarify what is meant by “visual inspection” and take into account that pulmonary metastases generally need a specific staining and a magnifying glass to be detected.

3) To correctly evaluate clock genes, it is relevant to record the time of sampling. Since the clock molecule levels (both mRNA and protein) change throughout the day, the differences found between evaluated groups could only respond to differences in the time of sampling. A mistake in the procedure of sampling could explain the contradiction of the increase of the Bmal1 at mRNA level but a decrease at protein level (Fig. 3). Authors should explain these methods in more detail. Additionally, to evaluate the clock in cell culture, it is necessary to synchronize the cells in order to put all in the same circadian phase, and, next, evaluate gene/protein levels at specific post-synchronization times. If it is possible, these evaluations should be performed at more than one time post-synchronization, or, at least, at the peak time. The authors should consider that, if the cells are not synchronized, the phase of the circadian cycle is different for each cell and thus, the levels of each clock gene are also different. In this case, the reporting data is limited to describing the average of the whole culture and it is impossible to know if the lack of Opn4 modified the levels of a specific clock gene (as Bmal1), and thus the clock machinery functioning, or only moved the phase of the circadian cycle but not the clock machinery functioning.

Page 12, lines 401-406: Authors say: “Although main clock proteins were not identified, several players (DBP, NDUFA9, PTGDS, TOP1, TOP2a, USP9x, and PP1CB) directly and indirectly involved in the positive loop regulation of the molecular clock (BMAL/CLOCK) were upregulated in Opn4WT (Supplementary Table S4). In the absence of OPN4, increased protein expression of HNRNPD, a negative regulator of the molecular clock, was identified (Supplementary Table 4)”. These data could indicate that the circadian clock has a different period (WT vs KO) but not be abolished in KO tumors. Authors should consider that the functioning of the molecular clock depends on the levels and postraductional modifications of several proteins, which together determine the amplitude and period of each circadian cycle as well as the phase of the rhythm. This fact makes it necessary to correctly evaluate the tissue sampling in order to improve the interpretation of the results.

4) Fig. 2: Authors should explain why the quantity of cells in G2/M phase is higher in mutant cells, since all remaining data indicates a decrease in proliferation rate. Why does this event not generate an increase in the number of cells in Fig. 2A? Similarly, authors say: “tumors with low

MITF expression display a reduced G1-S / G2-M ratio, indicating a higher proportion of cells in G2-M" (Fig. S5 C, page 9, lines 303-304), it seems to be contradictory that an increase of cells in G2-M phase (which supposes higher proliferation) in the better prognosis samples (low MITF levels). Authors should explain these statements.

In addition, a deeper discussion about the role of molecules involved in DNA damage response would also improve the manuscript.

5) Fig. S5: Since data previously shown and/or mentioned indicates a relation between MITF expression and cell proliferation, it is not clear why in the "MITF high" sample group there are fewer metastatic melanomas (Fig. S5 B).

In addition, it would be interesting to explore the expression levels of clock genes in the RNA-seq analyses.

6) Pages 14-15, lines 474-485: Authors discuss the role of the guanylyl cyclase activity in the decrease of tumor growth in KO mice. However, they do not mention the guanylyl cyclase inhibitor shown in Fig. 4 E. Since this molecule is reduced in these mice, it is relevant to include it in this discussion.

7) Page 19, lines 620-622: Authors say: "Rpl37a was used to normalize the expression values of the genes of interest in the in vitro and in vivo assays". In order to analyse clock gene expression, it is relevant to evaluate if the housekeeping gene used to normalize has not a circadian behaviour (change throughout the day). A common gene used for that is hprt.

8) In order to correctly analyze the flow cytometry assays, it is needed to evaluate the corresponding isotype controls. It could be shown in a supplementary figure.

9) Page 20, lines 636-637: Authors say: "Drugs and vehicles remained in dishes until the end of the experiment". To perform circadian determination, drugs used to synchronize cells are withdrawn from the culture (the culture medium is replaced by drug-free medium) in order to not induce chronic effects. Author should explain why they performed these experiments in this way.

Minor concerns:

1) Figs. 3 and 5: It is not clear why the result of MITF is shown in the middle of the results of Bmal1 and Per1. Since this molecule is not related with the circadian clock, it is recommended to show it at the end of figure or in a separate figure. There is a similar situation in the Result and Discussion sections when authors describe this data.

2) Pages 8-9, lines 292-294: Authors say: "Indeed, reduced Mitf expression (gene and protein) associated with a reduction at the gene level of important cell cycle regulators suggest that Opn4KO malignant melanocytes continue to exhibit impaired cell cycle progression in vivo." Please explain why authors suppose that the Opn4KO malignant melanocytes continue to exhibit impaired cell cycle progression, since all measured variables showed less proliferative phenotype.

3) Page 2-3, lines 109-111: Authors say: "Taken altogether, we provide evidence, for the first time, that OPN4 can act as a tumor suppressor gene in melanoma." This is confusing since the lack of this molecule reduces the tumor growth rate.

Page 13, lines 417-419: Similarly, authors say: "We provided evidence that a light- and thermo-sensing protein, OPN4, whose role as an important light sensor responsible for circadian entrainment has been well established 69, 70, plays a tumor-suppressing role in melanoma." It is relevant to clarify these statements as it seems to be context-dependent, as is mentioned later in line 433.

4) Page 15, lines 492-494: Authors say: "Based on these findings, one may suggest that OPN4 may interact with BMAL1 at the promoter region of MITF in a yet elusive mechanism" This data needs to be more exhaustively discussed. Has a protein-protein interaction of OPN4 and Bmal1 or the presence of OPN4 in the complex Bmal1-promoter been reported?

- 5) The phrase "oncogene gene" in the title seems redundant.
- 6) Line 47: The initials "CM" should be clarified.
- 7) Fig S2-B: the title says TAM-Opn4WT, I suppose it is a mistake and it should be TAM-Opn4KO.

Reviewer #1 (Remarks to the Author):

In this manuscript, the authors examine the effects of knocking out Opn4 on melanoma cell proliferation in vitro and in vivo. They find that loss of Opn4 is associated with reduced proliferation and tumor growth, which they found to be correlated with a number of changes in immune responses and Mitf signaling. Strengths of this manuscript include the relevance to melanoma (a growing and deadly problem) and the characterization of Opn4, for which there is little information currently available for its impact in cancer development. Though the authors have previously published work on Opn4 in normal melanocytes, the current manuscript is focused on metastatic melanoma cells. Knockout of Opn4 in these two model cell systems leads to different impacts on cell proliferation, which is intriguing, but not examined here.

Additional information can be provided about the generation of the Opn4 KO B16-F10 cells. For example, do the authors have any data showing the Opn4 is actually knocked out in their cell line (loss of protein by western blotting, for example)? Was a single KO clone used in all of their studies? Or, did they validate any findings on a second KO clone? Have the authors tried re-expressing Opn4 in their KO cell line to rescue the effects (and prove that the changes in the cell are due to Opn4 KO and not some other non-specific effect of using CRISPR)? Any information the authors can provide will help readers better understand the authors findings and potential limitations.

Authors' reply

We thank the reviewer for his/her comments on our manuscript.

The re-expression of OPN4 in *Opn4*KO cells was considered during the development of this project. However, we decided to focus our resources on the evaluation of the consequences of OPN4 knockout. We are certain that, although providing evidence for OPN4 rescue would benefit the manuscript and its findings, at this current moment due to funding constraints and limited staff due to the pandemic, we are unable to provide these findings. Considering these limitations, we have added a statement in the discussion about the limitations of our study as it now reads:

Lines 509 – 512: “Our findings are suggestive that OPN4 acts as an oncogene, but additional experiments using human melanoma cell lines and different approaches, i.e.,

knockdown, knockout, and/or rescue strategies are required to fully establish the OPN4 role as an oncogene.”

As requested by the reviewer detailed information about the generation of *Opn4*^{KO} cells are provided in the manuscript and it reads:

Lines 84 – 91: *“In this process, three clones that exhibited no ultraviolet A radiation-induced pigmentation and apoptosis responses were identified. B16-F10 Opn4KO clone 16 was chosen and Sanger sequencing of the CRISPR edited region showed alteration in the coding sequencing that led to the loss of function. Immunocytochemistry of OPN4 revealed increased protein presence in a region capping the nucleus, thus suggestive of protein retention likely due to altered protein structure²⁵. In this study, B16-F10 Opn4KO clone 16 was chosen and used in the next steps.”*

Minor issues:

1) I would suggest some editing for English and grammar. Some examples:

a) Title: "...novel oncogene gene..." Saying 'gene' twice is not necessary

b) Line 60: Mitf, not Mift

Authors' reply

Changes were made and the manuscript was proofread by an English-speaking expert.

Reviewer #3 (Remarks to the Author):

The manuscript “Melanopsin (OPN4) is a novel oncogene gene in cutaneous melanoma: evidence from an experimental study” describes the role of OPN4 in the tumoral context. Authors used different in vitro and in vivo strategies in both humans and a murine model. The silencing of OPN4 reduced the tumor growth rate and the level of molecules and mechanisms involved in cell proliferation, thus, it is considered an oncogene. In addition, the work also describes the behavior of immune variables as these mechanisms regulate the outcome of this pathology. The research was very well conducted but there are some specific explanations and analysis that could improve the manuscript.

Authors' reply

We thank the reviewer for his/her time invested in evaluating our manuscript.

1. Mayor concerns:

Page 4: lines 130-135: Authors say: “we inoculated C57Bl/6J mice, kept in thermoneutrality ($30 \pm 1^\circ\text{C}$), with B16-F10Opn4WT or B16-F10Opn4KO cells. It has been previously shown that tumor growth is significantly faster in mice kept at temperatures below thermoneutrality than at thermoneutrality 26. Indeed, tumor growth was only measurable from the 13th day onwards unlike previously shown in B16- F10Opn4WT inoculated mice kept at 22°C 19”. The habitual temperature to maintain mice is around 22°C , why do authors consider higher temperatures (30°C) normal? On the other hand, tumors tend to grow faster at higher temperatures. Authors should explain these protocols in detail in the methods section.

Authors' reply

As requested, additional information has been provided:

Lines 92 – 98: “To evaluate whether OPN4 would impact tumor development, we inoculated C57Bl/6J mice, kept in thermoneutrality ($30 \pm 1^\circ\text{C}$), with B16-F10 Opn4^{WT} or B16-F10 Opn4^{KO} cells. At temperatures between 29 and 31°C , mice do not activate thermogenesis to sustain core body temperature. In fact, mice kept below thermoneutrality are considered cold-stressed due to increased energy requirements to sustain core body temperature ²⁸. Therefore, to

avoid confounding factors caused by cold stress, mice were kept in their thermal-neutral temperature”

Lines 512 – 514: *“One must consider that our study used mice kept at thermoneutrality, and thus, comparisons with other studies that used temperature lower than the thermoneutral, should be made with caution.”*

2. Mayor concerns:

1) Line 174: Author said: “...decreased gene expression of both pro- (Il-1 β and Tgf- β) and anti-inflammatory (Il-6 and Il-10) players...” Tgf- β is an anti-inflammatory cytokine, and il-6, in spite of its pleiotropic functions, is classified as pro-inflammatory cytokines since it is secreted during inflammation together with il-1 β and TNF- α . Author should correct it.

Authors’ reply

We apologize for this mistake, which has been fixed.

Lines 138 – 142: *“Intriguingly, decreased gene expression of both pro- (Il-1 β and Il-6) and anti-inflammatory (Il-10 and Tgf- β) players, as well as T CD8+ dependent effector function genes such as granzyme (Gzma) and perforin (Prf1) in TME of Opn4^{KO} was found when compared to Opn4^{WT} tumor-bearing mice (Fig. 1 O – T)”*

3. Mayor concerns:

Page 10, lines 351-356: Authors say: “These proteins were grouped as “negative regulation of interleukin-17 secretion” in Opn4WT tumors. Indeed, the presence of Th17 cells in TME is a common feature in many cancers 52, 53 and the evasion of Th17 cells constitutes an immune evasion mechanism 54. Therefore, the suppressor activity of PARP1 and OTUD5 in IL-17 production in TME could be associated with the decreased TIL levels found in Opn4WT tumors (Fig. 1).” It is confusing that authors should designate a pro-tumoral (evasion prone) role to Th17 cells when these cells decrease in WT mice in which the tumors grow faster. Moreover, the role of pro-inflammatory immune mechanisms, as in those mediated by Th1, Th17 or M1 cells, in the progression/regression of tumors depends on the tumor type. A more in-depth and specific interpretation and discussion is needed about this point.

Additionally, this result and the cytokine levels shown in Fig. 1 could be collectively interpreted since Tgf- β and il-6 participate in the differentiation of CD4 naive to Th17 cells (considering the differences between human and mice in this process).

Authors' reply

We are grateful for the input made by the reviewer. We would like to clarify that we suggested a possible mechanism to explain the reduced TILs levels in OPN4^{WT} tumors. The presence of PARP1 and DUBA were only identified in the OPN4^{WT} proteome, and since these proteins negatively regulate TH17 cells, we suggested that mechanism to reduce TILs in OPN4^{WT} tumors. However, as mentioned by the reviewer, increased levels of *Tgf-b* and *Il6* are expected to result in higher levels of Th17, which could benefit tumor growth. Since Th17 population was not evaluated in our study, we have raised this possibility but refrained ourselves to associate the increased tumor growth with Th17 signaling.

It now reads: **Line 323 – 329:** *“On the other hand, increased levels of Tgf- β and Il-6 in Opn4WT tumors (Fig. 1) could result in CD4+ naïve differentiation into Th17 cells^{57,58}. One might consider that increased Th17 cells would lead to enhanced tumor growth as supported by experimental evidence. However, fact, anti-tumoral effects of Th17 in melanoma have also been described⁵⁹. However, the CD4+ Th17 population was not evaluated in this study and further investigation is required”*

4. Mayor concerns:

Page 11, lines 385-388: Similarly, authors say: “Furthermore, significantly increased levels of Complement factor B (CFB) in Opn4WT tumors compared to Opn4KO, and with exclusively identified C5 and C4b proteins in Opn4WT proteome, could be associated with suppression of the immune response in Opn4WT melanoma by inhibiting TIL function 68.” Since the complement mechanism is also related to the inflammation process, this association should be discussed in more detail.

Authors' reply

We have provided a better description and discussion of the increased levels of complement proteins in OPN4WT tumors and their role in inflammation and tumor progression.

It now reads **lines 360 – 363**: “It is known that complement system activation in cancer is linked with a higher inflammatory TME and melanoma progression^{73,74}. Therefore, these data together suggest a higher inflammatory TME, which could explain the higher tumor growth of *Opn4WT* tumors.”

Lines 491 – 508: “Evidence of a less inflammatory TME in *Opn4KO* tumors was found as reduced complement expression and gene expression of inflammatory cytokines were identified in *Opn4KO* tumors. Reduced expression of GTPases has been linked with slower cell cycle progression⁸⁸, which is in line with the slower *Opn4KO* proliferation. Moreover, the higher proliferative capacity of *Opn4WT* tumors can be associated with the exclusive presence of DNA repair proteins (*PARP1*, *RPA2*, *APEX1*, and *XRCC6*) and DNA replication (*RCF2* and *TOP1*). Increased DNA repair capacity has been related with more aggressive cancer development and treatment resistance in melanoma⁸⁹⁻⁹². We previously identified a link between higher expression of *BMAL1* and lower base excision repair (BER) score. Indeed, such association was found in our experimental model as the absence of *OPN4* resulted in elevated *BMAL1* protein expression as well as reduction/absence of BER-related proteins (*PARP1*, *RPA2*, *APEX1*⁹³⁻⁹⁵; uniprot database). Since these proteins were only identified in *Opn4WT* tumors, we suggest a possible reduction of DNA repair via decreased BER activity. It may be possible that these findings are linked with reduced *MITF* signaling in the absence of *OPN4* via an elusive mechanism. Although *MITF* was not detected in the proteomics, we found that classical *MITF* targets such as *DCT*, *TYR*, and *TYRP1* were less expressed in *Opn4KO* tumors”

5. Mayor concerns:

It is relevant to consider the complexity of the immune response in the tumor context and the particularities observed in each specific tumor. A more complete and robust discussion about the immune role is needed.

Authors’ reply

We agree with the reviewer’s criticism. In this new version, we have focused on the immune system interaction only in melanoma, thus avoiding generalization. The response to this comment can be appreciated in the other replies.

6. Mayor concerns:

2) Since the B16-F10 tumor cell line induces pulmonary metastases and the number of tumor cells inoculated was very high (2.106 cells), authors should explain their thinking about the lack of metastasis in this case and include an explanation of the procedures by means of which they evaluated metastases. In this regard, authors say: “After euthanasia, every animal was visually inspected, and no metastasis was found”, please clarify what is meant by "visual inspection" and take into account that pulmonary metastases generally need a specific staining and a magnifying glass to be detected.

Authors' reply

In our experimental model of subcutaneous B16-F10 inoculation in animals, either kept at 22 or 30°C, no visible nodules of B16-F10 metastasis have been found in the organs. We agree with the reviewer's criticism as it is possible to have visually unnoticed metastasis nodules.

Therefore, we have rephrased our statement and it reads now:

Lines 562 – 564: “*After euthanasia, every animal was visually inspected, and no visible metastasis nodules were found.*”

7. Mayor concerns:

3) To correctly evaluate clock genes, it is relevant to record the time of sampling. Since the clock molecule levels (both mRNA and protein) change throughout the day, the differences found between evaluated groups could only respond to differences in the time of sampling. A mistake in the procedure of sampling could explain the contradiction of the increase of the *Bmal1* at mRNA level but a decrease at protein level (Fig. 3). Authors should explain these methods in more detail.

Authors' reply

We thank the reviewer for this important comment. Sample collection followed a strict protocol and now it is properly mentioned. The sampling of the *in vivo* experiments took place between ZT 2 and 3. All the *in vitro* experiments sampling, except for the XTT assay, occurred 96 h after cell seeding. Therefore, it is our understanding that sampling time cannot explain the discrepancy between *Bmal1* gene and protein.

Proper information is now provided.

Lines 561 – 562: “Animals were CO₂ euthanized 25 days after inoculation, and death was assured by cervical dislocation, between 9 and 10 am (Zeitgeber time 2 – 3).”

We have also stated one possibility that could explain the lack of consistency between *Bmal1* gene and protein.

Lines 201 – 204: “The lack of concordance between *Bmal1* gene and protein data might be attributed to post-translational modifications of *BMAL1* protein (Hirano et al., 2016), which can result in a longer protein half-time.”

8. Mayor concerns:

Additionally, to evaluate the clock in cell culture, it is necessary to synchronize the cells in order to put all in the same circadian phase, and, next, evaluate gene/protein levels at specific post-synchronization times. If it is possible, these evaluations should be performed at more than one time post-synchronization, or, at least, at the peak time. The authors should consider that, if the cells are not synchronized, the phase of the circadian cycle is different for each cell and thus, the levels of each clock gene are also different. In this case, the reporting data is limited to describing the average of the whole culture and it is impossible to know if the lack of *Opn4* modified the levels of a specific clock gene (as *Bmal1*), and thus the clock machinery functioning, or only moved the phase of the circadian cycle but not the clock machinery functioning.

Authors' reply

Our circadian analyses were based on the *Per1::Luc* bioluminescence assay while qPCR and flow cytometry was only performed in a single timepoint. We would like to clarify that in our hands, the B16-F10 cells showed a weak entrainment capacity in response to different strategies compared to normal melanocytes. Moreover, in-phase clock gene expression of *Per1* and *Bmal1* has been previously reported, which suggests a disrupted clock machinery in B16-F10 (PMID: 26947915; PMID: 27535239).

Considering the reduced capacity of cell synchronization, our strategy was not designed to evaluate the synchronization and/or clock rhythmicity in the presence or absence of *OPN4*, but rather to evaluate the acute response of this machinery to dexamethasone and forskolin. To that

end, we left the drugs until the end of the experiment, which unfortunately results in the over-activation of some pathways. We are aware that such a strategy does not allow us to validate the circadian clock response in the presence and absence of OPN4. In addition, we decided to keep this experimental setup similar to the pharmacological assays, in which drugs were also kept in the media throughout the experiment.

Proper alterations were made as they read now:

Lines 209 – 211: *“Since B16-F10 cells show a reduced synchronization capacity to different methods^{18,19,44}, we focused on the evaluation of the acute clock response in this assay”*

Lines 221 – 224: *“Interestingly, Opn4^{KO} malignant cells were less responsive to both treatments (Fig. 3 E - H). These data suggest that the acute molecular clock response is impaired in the absence of Opn4. Further circadian-related experiments are needed to clarify the impact of OPN4 in the regulation of the circadian clock function.”*

Lines 444 – 449: *“The experimental design chosen in this study focused on the evaluation of the acute response of the molecular clock and not on the rhythmicity aspect in the absence of Opn4 since the B16-F10 circadian clock has previously shown weak responses to synchronizing agents^{18,19}. Therefore, our experimental data are insufficient to evaluate the alterations that the absence of Opn4 may cause to the circadian clock function”*

9. Mayor concerns:

Page 12, lines 401-406: Authors say: “Although main clock proteins were not identified, several players (DBP, NDUFA9, PTGDS, TOP1, TOP2a, USP9x, and PP1CB) directly and indirectly involved in the positive loop regulation of the molecular clock (BMAL/CLOCK) were upregulated in Opn4WT (Supplementary Table S4). In the absence of OPN4, increased protein expression of HNRNPD, a negative regulator of the molecular clock, was identified (Supplementary Table 4)”. These data could indicate that the circadian clock has a different period (WT vs KO) but not be abolished in KO tumors. Authors should consider that the functioning of the molecular clock depends on the levels and postraductional modifications of several proteins, which together determine the amplitude and period of each circadian cycle as well as the phase of the rhythm. This fact makes it necessary to correctly evaluate the tissue sampling in order to improve the interpretation of the results.

Authors' reply

We agree with the reviewer's comments and how the sampling time would affect the conclusions of our findings. However, as mentioned above, sampling time for all experiments followed a strict protocol, thus ruling out time as a variable in our analyses.

We have clarified this in the M&M section and also in the discussion.

Lines 514 – 517: *“With regards to the circadian aspects, our in vitro and in vivo sampling collection took place within a narrow time interval, thus, ruling out the influence of time in our analyses. However, additional circadian experiments are necessary to fully establish the impact of OPN4 in the regulation of the circadian clock”*

Lines 561 – 562: *“Animals were CO₂-euthanized 25 days after inoculation, and death was assured by cervical dislocation between 9 and 10 am (Zeitgeber time 2 – 3)”*.

10. Mayor concerns:

4) Fig. 2: Authors should explain why the quantity of cells in G2/M phase is higher in mutant cells, since all remaining data indicates a decrease in proliferation rate. Why does this event not generate an increase in the number of cells in Fig. 2A? Similarly, authors say: “tumors with low MITF expression display a reduced G1-S / G2-M ratio, indicating a higher proportion of cells in G2-M” (Fig. S5 C, page 9, lines 303-304), it seems to be contradictory that an increase of cells in G2-M phase (which supposes higher proliferation) in the better prognosis samples (low MITF levels). Authors should explain these statements.

Authors' reply

Our working hypothesis is that *Opn4*^{KO} cells display an impairment in cell cycle progression due to increased G2/M retention. Our interpretation is that increased G2/M cells results in cells arrested during cycle in this phase, and thus in agreement with less proliferation observed in Fig 1 and Fig 2. Our analyses not only focused on the cell cycle assay but also on other techniques to create a link between the lower proliferation rate and increased G2/M retention of *Opn4*^{KO}. For instance, reduced *Cyclin F* and *Check1* expression was found in *Opn4*^{KO} tumor samples, and proteomics analyses showed higher expression of G2/M proteins in the same samples. In *Opn4*^{WT} tumors, G1/S proteins were present at a higher level. Importantly,

lower GTPase activity was also found in the absence of OPN4, which has been linked with reduced proliferation.

Therefore, we suggest a slower proliferative capacity in the absence of OPN4 via a process that is not fully characterized, which is now better discussed in the manuscript.

Lines 434 – 441 – *“The in vitro data demonstrated that reduced cell proliferation is associated with reduced cell metabolism and proliferative capacity. Of note, cell cycle progression was also impaired in the absence of Opn4, as these cells exhibited a decrease in G1 and S as well as an increase in the G2/M phases. These increased levels of G2/M could in fact represent a point of cell cycle arrest, related to the absence of Opn4^{KO}. Therefore, our findings suggest that Opn4^{KO} malignant melanocytes display an impaired capacity to fully start a new cell cycle. Important cell cycle regulators were also affected at the mRNA level, which collectively confirm a slower proliferation phenotype”*

Upon analyzing the TCGA dataset, we found evidence that low expressing MITF tumors also show a higher expression of G2/M related genes. Because only gene expression is available in TCGA dataset, we cannot infer the growth properties of the tumors, but it could be inferred based on our experimental data. Importantly, MITF expression directly correlates with cell proliferation, and thus, low MITF tumors are known to display decreased proliferation (PMID: 28263292)

11. Mayor concerns:

In addition, a deeper discussion about the role of molecules involved in DNA damage response would also improve the manuscript.

11.1. Authors' reply

As requested, a statement has been included:

Line 495 – 499: *“Moreover, the higher proliferative capacity of Opn4^{WT} tumors can be associated with the exclusive presence of DNA repair proteins (PARP1, RPA2, APEX1, and XRCC6) and DNA replication (RCF2 and TOP1). Increased DNA repair capacity has been related with more aggressive cancer development and treatment resistance in melanoma⁸⁹⁻⁹²”*

12. Mayor concerns:

5) Fig. S5: Since data previously shown and/or mentioned indicates a relation between MITF expression and cell proliferation, it is not clear why in the “MITF high” sample group there are fewer metastatic melanomas (Fig. S5 B). In addition, it would be interesting to explore the expression levels of clock genes in the RNA-seq analyses.

Authors' reply

In human melanoma lower MITF expression has been classically associated with a stem cell-like phenotype that is more invasive (metastatic). Conversely, high MITF expression is found in primary and less invasive tumors (PMID: 28263292). The role of melanoma clock genes in carcinogenesis and therapeutics has been investigated by our group (PMID: 29946530). Only *Bmal1* was shown to play a significant role in the melanoma carcinogenic process. This paper has been cited in our current manuscript.

13. Mayor concerns:

6) Pages 14-15, lines 474-485: Authors discuss the role of the guanylyl cyclase activity in the decrease of tumor growth in KO mice. However, they do not mention the guanylyl cyclase inhibitor shown in Fig. 4 E. Since this molecule is reduced in these mice, it is relevant to include it in this discussion.

Authors' reply

As requested, such information has been added as it now reads:

Lines 463 – 465: *“Within this line, our in vitro pharmacological guanylyl cyclase inhibition was effective only in the presence of OPN4, thus suggesting that this enzyme activity in *Opn4*^{KO} is at an already low level.”*

14. Mayor concerns:

7) Page 19, lines 620-622: Authors say: “Rpl37a was used to normalize the expression values of the genes of interest in the in vitro and in vivo assays”. In order to analyse clock gene expression, it is relevant to evaluate if the housekeeping gene used to normalize has not a circadian behaviour (change throughout the day). A common gene used for that is *hprt*.

Authors' reply

For Melan-a and B16-F10 cells, *Rpl37a* showed a very good profile for a housekeeping gene, i.e., exhibited robust stability ($SD < 1$ across all samples). *Eef1a* has also been shown to be a good normalizer gene for skin and its cells, in our experience. Therefore, the gene expression differences cannot be attributed to the normalizer, but rather to the expression of the target gene. In this study, the circadian profile of gene expression was not investigated.

Line 629 – 630: “*Rpl37a* showed a robust and stable expression across samples ($SD < 1$).”

15. Mayor concerns:

8) In order to correctly analyze the flow cytometry assays, it is needed to evaluate the corresponding isotype controls. It could be shown in a supplementary figure.

Authors' reply

Representative negative controls of all flow cytometry experiments are found in Fig S2, as requested.

Line 681 – 682: “*Representative negative controls of all flow cytometry experiments can be found in Figure S2 A-I.*”

16. Mayor concerns:

9) Page 20, lines 636-637: Authors say: “Drugs and vehicles remained in dishes until the end of the experiment”. To perform circadian determination, drugs used to synchronize cells are withdrawn from the culture (the culture medium is replaced by drug-free medium) in order to not induce chronic effects. Author should explain why they performed these experiments in this way.

Authors' reply

We thank the reviewer for this important comment. We agree that this strategy would not be appropriate to evaluate the circadian regulation of *OPN4* *in vitro*. However, the circadian oscillation of *Opn4* was not addressed. We kindly request the reviewer to see our answer to question 8 as it also covers the point raised here.

17. Minor concerns:

1) Figs. 3 and 5: It is not clear why the result of MITF is shown in the middle of the results of Bmal1 and Per1. Since this molecule is not related with the circadian clock, it is recommended to show it at the end of figure or in a separate figure. There is a similar situation in the Result and Discussion sections when authors describe this data.

Authors' reply

An interesting link between MITF and the circadian clock has been recently described by Gaddameedhi's lab, which was fully disclosed in our discussion. Since the absence of *Opn4* led to a reduction in MITF expression, we hypothesized that an interaction between OPN4, MITF, and BMAL1 may exist, thus justifying the inclusion of MITF in the clock data description. Since *Mitf* gene expression is considered an important piece of data, we decided to place it in the main figure and not in the supplements.

We have changed Figure 5 as requested.

18. Minor concerns:

2) Pages 8-9, lines 292-294: Authors say: "Indeed, reduced *Mitf* expression (gene and protein) associated with a reduction at the gene level of important cell cycle regulators suggest that *Opn4*KO malignant melanocytes continue to exhibit impaired cell cycle progression in vivo." Please explain why authors suppose that the *Opn4*KO malignant melanocytes continue to exhibit impaired cell cycle progression, since all measured variables showed less proliferative phenotype.

Authors' reply

We agree with the reviewer. Our data support a less proliferative phenotype in the *in vivo* tumors. Therefore, a modification was made, and it now reads:

Line 263 – 265: "*Indeed, reduced Mitf expression (gene and protein) associated with a reduction at the gene level of important cell cycle regulators suggest that Opn4^{KO} malignant melanocytes continue to exhibit impaired proliferation when inoculated in vivo.*"

19. Minor concerns:

3) Page 2-3, lines 109-111: Authors say: “Taken altogether, we provide evidence, for the first time, that OPN4 can act as a tumor suppressor gene in melanoma.” This is confusing since the lack of this molecule reduces the tumor growth rate.

Authors’ reply

We apologize for this mistake. It is now corrected, and it reads:

Line 518 – 519: *“Taken altogether, our data add a novel layer of complexity to the opsin realm as we provide evidence that OPN4 can be a tumor oncogene in melanoma.”*

20. Minor concerns:

Page 13, lines 417-419: Similarly, authors say: “We provided evidence that a light- and thermo-sensing protein, OPN4, whose role as an important light sensor responsible for circadian entrainment has been well established 69, 70, plays a tumor-suppressing role in melanoma.” It is relevant to clarify these statements as it seems to be context-dependent, as is mentioned later in line 433.

Authors’ reply

We apologize for the mistake, and it now reads:

Line 392 – 394: *“We provided evidence that a light- and thermo-sensing protein, OPN4, whose role as an important light sensor responsible for circadian entrainment has been well established^{75,76}, plays a pro-tumoral role in melanoma.”*

21. Minor concerns:

4) Page 15, lines 492-494: Authors say: “Based on these findings, one may suggest that OPN4 may interact with BMAL1 at the promoter region of MITF in a yet elusive mechanism” This data needs to be more exhaustively discussed. Has a protein-protein interaction of OPN4 and Bmal1 or the presence of OPN4 in the complex Bmal1-promoter been reported?

Authors’ reply

To the best of our knowledge, no study has evaluated whether OPN4 displays DNA binding activity or whether it associates with BMAL1-DNA bindings proteins. However, our suggestion emerges from our data and from Gaddameedhi’s study.

We reported that the absence of OPN4 results in reduced MITF while it has been shown that MITF displays a rhythmic expression as it contains E-box sequences in its promoter region.

Therefore, we suggested that OPN4 could participate in this process either by promoting alteration in the promoter region or downstream effects such as mRNA and/or protein degradation.

The new statement now reads as:

Line 475 – 478: *“We suggest that OPN4 could participate in the circadian regulation of MITF either by interacting with BMAL1 at the DNA level and/or via downstream pathways that lead to degradation of mRNA and/or protein. Further research is needed to clarify this possible interaction”*

22. Minor concerns:

- 5) The phrase “oncogene gene” in the title seems redundant
- 6) Line 47: The initials “CM” should be clarified.
- 7) Fig S2-B: the title says TAM-Opn4WT, I suppose it is a mistake and it should be TAM-Opn4KO.

Authors’ reply

Corrections were made in the revised version of the manuscript.

REVIEWERS' COMMENTS:

Reviewer #1 (Remarks to the Author):

The authors have addressed my concerns.

Reviewer #3 (Remarks to the Author):

The manuscript entitled "Melanopsin (OPN4) is a novel oncogene in cutaneous melanoma" has been greatly improved. The authors have adequately corrected and/or explained the points mentioned in the previous revision. I suggest to published it.